# Global Observations of Aerosol Indirect Effects from Marine Liquid Clouds

Casey J. Wall[1], Trude Storelvmo[1,2], Anna Possner[3]

[1]Department of Geosciences, University of Oslo, Oslo, 0371, Norway
[2]Nord University, Bodø, 8026, Norway
[3]Institute for Atmospheric and Environmental Sciences, Goethe University Frankfurt, Frankfurt, 60438, Germany

*Correspondence to*: Casey J. Wall (c.j.wall@geo.uio.no)

**Abstract.** Interactions between aerosols and liquid clouds are one of the largest sources of uncertainty in the historical radiative forcing of climate. One widely shared goal to reduce this uncertainty is to decompose radiative anomalies arising
from aerosol-cloud interactions into components associated with changes in cloud-droplet number concentration (Twomey effect), liquid-water-path adjustments, and cloud-fraction adjustments. However, there has not been a quantitative foundation for simultaneously estimating these components with global satellite observations. Here we present a method for assessing shortwave radiative flux anomalies from the Twomey effect and cloud adjustments over ocean between 55°S and 55°N. We find that larger aerosol concentrations are associated with widespread cloud brightening from the Twomey effect, a positive
radiative adjustment from decreasing liquid water path in subtropical stratocumulus regions, and a negative radiative adjustment from increasing cloud fraction in the subtropics and midlatitudes. The Twomey effect and total cloud adjustment contribute $-0.77 \pm 0.25$ W m$^{-2}$ and $-1.02 \pm 0.43$ W m$^{-2}$, respectively, to the effective radiative forcing since 1850 over the domain (95% confidence). Our findings reduce uncertainty in these components of aerosol forcing and suggest that cloud adjustments make a larger contribution to the forcing than is commonly believed.

## 1 Introduction

Changes in aerosol concentrations over the industrial era have modified clouds and perturbed the global radiation balance at the top of the atmosphere (Raghuraman et al., 2021; Kramer et al., 2021). The radiative flux perturbation resulting from these cloud changes, known as the effective radiative forcing from aerosol-cloud interactions (ERF$_{aci}$), is estimated to be $-0.84 \pm 0.61$ W m$^{-2}$ between 1750 and 2019 (90% confidence interval (CI) from Forster et al. (2021)). ERF$_{aci}$ is much
more uncertain than the positive radiative forcing from carbon dioxide changes ($+2.16 \pm 0.26$ W m$^{-2}$), meaning that ERF$_{aci}$ offsets a potentially large but highly uncertain portion of historical greenhouse-gas forcing. Reducing this uncertainty would improve assessments of climate sensitivity and committed future warming (Matthews and Zickfeld, 2012; Mauritsen and Pincus, 2017; Sherwood et al., 2020; Watson-Parris and Smith, 2022).

An extension of these forcing estimates is to characterize how changes in different cloud properties contribute to
ERF$_{aci}$, thereby providing insight into the relative importance of different processes. For instance, as cloud condensation

nuclei (CCN) become more abundant, liquid clouds typically form smaller but more numerous droplets. The change in cloud droplet effective radius ($r_e$) and number concentration ($N_d$) directly increases cloud optical thickness – a mechanism known as the Twomey effect (Twomey, 1977). The reduction of cloud droplet size can also enhance evaporation or reduce precipitation, causing adjustments in cloud thickness, lifetime, or morphology (Albrecht, 1989; Pincus and Baker, 1994;

Rosenfeld et al., 2006; Bretherton et al., 2007). Separating the radiative impacts of the Twomey effect and cloud adjustments is thus an important step towards understanding the causes of ERF$_{aci}$.

Recent community assessments find that the components of ERF$_{aci}$ all have considerable uncertainty. The Sixth Assessment Report of the Intergovernmental Panel on Climate Change (IPCC) estimates that the Twomey effect is $-0.7 \pm 0.5$ W m$^{-2}$, the adjustment of liquid water path (LWP) is $+0.2 \pm 0.2$ W m$^{-2}$, and the adjustment of liquid-cloud fraction is

$-0.5 \pm 0.4$ W m$^{-2}$ (90% CIs for forcing between 1750 and 2014) (Forster et al., 2021). Another assessment by the World Climate Research Programme reports even larger uncertainties (Bellouin et al., 2020). In particular, they find that the cloud-fraction adjustment is especially uncertain: It could be negligibly small or large enough to offset most of the historical carbon-dioxide forcing.

Constraints from satellite observations offer a path toward reducing this uncertainty, but in practice it has been

difficult to isolate relationships between aerosols and radiative anomalies caused by changes in individual cloud properties (Feingold et al., 2022). Furthermore, previous global observational studies that attempt to quantify these relationships used separate methods for estimating the Twomey effect, LWP adjustment, and cloud-fraction adjustment, so their estimates may suffer from limitations that differ from one ERF$_{aci}$ component to another (Forster et al., 2021). This complicates efforts to rigorously compare the Twomey effect and cloud adjustments. Here we address these challenges by adapting techniques

from the cloud-feedback literature. We develop a cloud radiative kernel that separates the radiative anomalies caused by changes in $r_e$, LWP, and liquid-cloud amount, and we relate each of these radiative anomalies to local aerosol concentrations and $N_d$. This facilitates an assessment of the Twomey effect and liquid-cloud adjustments over the global ocean.

## 2 Data and Methods

### 2.1 Satellite Data, Reanalysis, and Climate Model Output

We analyze monthly gridded satellite observations from 2003 through 2020 obtained from the Moderate Imaging Spectroradiometer (MODIS) MCD06COSP dataset version 6.2.0 (Pincus et al., 2023). This dataset combines observations from MODIS instruments on the Aqua and Terra satellites. Our primary unit of analysis is a joint histogram of pixel counts for liquid-topped clouds partitioned by $r_e$ and LWP. Histogram counts are normalized by the number of all valid pixels in the grid box and then multiplied by 100 to convert the units to cloud fraction (Fig. 1a). These histograms represent the fractional

occurrence of liquid-topped clouds that are exposed to space, hence they do not include cases where liquid cloud is obscured by overlying ice. LWP is defined by the vertical integral of cloud liquid water mass per unit area, and $r_e$ is defined by the

ratio of the third and second moments of the cloud-droplet radius distribution. These two variables are estimated with the MODIS 3.7 $\mu$m retrieval algorithm (Platnick et al., 2017).

We also use daily gridded $N_d$ estimates from MODIS (Gryspeerdt et al., 2022) and monthly gridded radiative flux retrievals from the Clouds and the Earth's Radiant Energy System (CERES) Energy Balanced and Filled ed. 4.1 and FluxByCldTyp ed. 4.1 datasets (Loeb et al., 2018; Sun et al., 2022). The MODIS $N_d$ estimates can be biased when $r_e$ is sufficiently small, when the cloud visible optical thickness is sufficiently small, or when three-dimensional radiative transfer effects contribute to the measured radiances (Grosvenor et al., 2018). To avoid these problematic cases, $N_d$ is estimated for single-layer liquid clouds that satisfy several conditions: (i) $r_e$ is larger than 4 $\mu$m, (ii) cloud optical thickness is larger than four, (iii) cloud fraction at 5 km resolution is larger than 0.9, (iv) the solar zenith angle is less than 65°, (v) the satellite viewing zenith angle is less than 55°, (vi) the sub-pixel heterogeneity index defined by Zhang and Platnick (2011) is less than 0.3, (vii) the MODIS estimate of $r_e$ is largest from the 3.7 $\mu$m retrieval algorithm, followed by the 2.1 $\mu$m retrieval algorithm and then the 1.6 $\mu$m retrieval algorithm, and (viii) cloud optical thickness is in the top 10% of values in 100 km × 100 km regions ("Z18 sampling" in Gryspeerdt et al. (2022)). The final condition preferentially selects the convective cores in cloudy scenes (Zhu et al., 2018). Most cloud droplets in shallow convective clouds form near the cloud base, so $N_d$ in convective cores depends on the CCN concentration in air that enters the cloud from below (Rosenfeld et al., 2019). Thus, $N_d$ in convective cores typically does not represent $N_d$ in the entire cloud, but it serves as an indicator of CCN concentration near cloud base. We denote the estimated cloud-droplet number concentration as $\widetilde{N}_d$ to distinguish it from the cloud-droplet number concentration in the entire cloud. For consistency with the MODIS cloud histograms, we use $\widetilde{N}_d$ estimates from the MODIS 3.7 $\mu$m retrieval algorithm, and we combine data from the Terra and Aqua satellites. Daily values of $\ln \widetilde{N}_d$ are averaged across the satellite platforms and over one-month intervals, weighted by the number of pixels with a valid $\widetilde{N}_d$ retrieval.

Monthly meteorological fields and the dry mass concentration of sulfate aerosol at 910 hPa, $s$, are obtained from MERRA-2 reanalysis (Gelaro et al., 2017; Randles et al., 2017). We consider sulfate aerosol because it dominates the anthropogenic influence on CCN (Charlson et al., 1992; Stevens, 2015), and we select data from 910 hPa rather than the surface because the 910 hPa level is a better indicator of aerosol concentration near cloud base (Painemal et al., 2017). The sulfate data are determined from bias-corrected observations of total aerosol optical depth from cloud-free pixels and simulations from a global model that treats the sources, sinks, and chemistry of sulfate and its precursor gases. The data assimilation accounts for aerosol swelling in humid environments and filters out pixels near clouds that are affected by retrieval bias (Randles et al., 2017). The main limitation of the sulfate data is that the total aerosol optical depth is constrained by observations, but aerosol species distributions and vertical profiles are not. These data provide an additional indicator of cloud-base CCN concentration that is independent of the MODIS estimates of $\widetilde{N}_d$.

Finally, we use output from historical simulations of 20 global climate models (GCMs) from the Coupled Model Intercomparison Project Phase 6 (CMIP6). The simulations are run from 1850 through 2014 with realistic emissions of

greenhouse gases, aerosols, and aerosol precursor gases. Sulfate mass concentration from the model output is converted to pressure coordinates and linearly interpolated to 910 hPa. The models are listed in Supplementary Table 1.

## 2.2 Quantifying Aerosol Indirect Effects

We first relate variability of aerosols to the radiative effects of liquid clouds – relationships we call "aerosol indirect effects." Our analysis begins with the MODIS joint histograms of liquid-cloud fraction, $C$, partitioned by $r_e$ and LWP (Fig.
1a). For a given latitude, longitude, and time, the shortwave (SW) radiative flux anomaly at the top of the atmosphere that is induced by liquid clouds, $R'$, is estimated according to

$$R' = \sum_{r=1}^{6} \sum_{l=1}^{7} C'_{rl} \frac{\partial R}{\partial C_{rl}},$$

where $r$ and $l$ represent the $r_e$ and LWP dimensions of the histogram and primes denote monthly anomalies relative to the local climatological seasonal cycle. The cloud radiative kernel, $\partial R / \partial C_{rl}$, represents the SW flux anomaly that would occur if
the cloud fraction $C_{rl}$ were to increase by 1% with all non-cloud factors fixed (Fig. 1b). The kernel is computed with the Rapid and Accurate Radiative Transfer Model for Global Climate Models (Clough et al., 2005) following a method similar to that of Zelinka et al. (2012). We then adapt a method of cloud-feedback analysis developed by Zelinka et al. (2013) to decompose $R'$ into contributions from different cloud properties:

$$R' = R'_{r_e} + R'_{\text{LWP}} + R'_{\text{CF}} + R'_{\text{res}}.$$

$R'_{r_e}$, $R'_{\text{LWP}}$, and $R'_{\text{CF}}$ are the SW flux anomalies caused by $r_e$, LWP, and cloud-amount anomalies, respectively – each computed with the other properties held fixed. We note that $R'_{r_e}$ is the radiative anomaly that is caused by variations in $r_e$ with fixed LWP, so it is equivalent to the radiative anomaly that is directly caused by variations in $N_d$. $R'_{\text{res}}$ is the residual of the decomposition. The radiative kernel and MODIS joint histograms reproduce monthly observations of $R'$ across the global ocean with a bias of about +4.6% (Appendix A). The methods for computing the kernel and decomposing $R'$ are described
in Appendix A and B.

To test robustness of the results, we make one set of $R'$ estimates in which only fully cloud-covered pixels are included in the histograms and a second set of estimates in which fully and partly cloud-covered pixels are both included. We refer to these cases as MODIS$_{\text{CLD}}$ and MODIS$_{\text{CLD+PCL}}$, respectively. The filter of the MODIS$_{\text{CLD}}$ case avoids retrieval biases that affect partly cloudy pixels, but it may introduce a sampling bias by excluding some cloud elements. The opposite
is true for MODIS$_{\text{CLD+PCL}}$. Both cases are presented to explore tradeoffs between the accuracy and completeness of the satellite cloud data.

We relate $R'$ to sulfate and cloud-droplet concentrations using cloud-controlling factor analysis (Scott et al., 2020; Myers et al., 2021). Our analysis closely follows the method of Wall et al. (2022, hereafter W22), except that we generalize their results by applying cloud-controlling factor analysis to $R'$ and each of its components. The cloud-controlling factor
method approximates $R'$ as a linear combination of seven local cloud-controlling factors $x_i$:

$$R' \approx \sum_{i=1}^{7} \frac{\partial R}{\partial x_i} x_i'.$$

The first six $x_i$ terms include sea surface temperature, estimated inversion strength at the top of the planetary boundary layer (Wood and Bretherton, 2006), low-level advection across a surface-temperature gradient, surface wind speed, relative humidity at 700 hPa, and vertical wind at 700 hPa. Collectively these terms represent all of the standard large-scale meteorological controls on liquid clouds in the marine boundary layer that have been proposed in the literature (Scott et al., 2020). The final $x_i$ term can be either $\ln s$ or $\ln \widetilde{N}_d$. All meteorological terms and $\ln s$ are calculated with MERRA-2 data and linearly interpolated to the native $1° \times 1°$ grid of MODIS. We then select ocean-covered grid boxes, remove the climatological seasonal cycle and least-squares linear trend from all variables in each grid box, and average the anomalies over a $5° \times 5°$ grid that spans 55°S to 55°N. For each ocean grid box, $R'$ is regressed against anomalies of the seven cloud-controlling factors using ordinary least-squares multilinear regression. Separate regressions are performed with $\ln s$ and $\ln \widetilde{N}_d$ as the final predictor. Thus, the regression coefficients $\partial R / \partial \ln s$ and $\partial R / \partial \ln \widetilde{N}_d$ represent the relationship between $R'$ and local anomalies of $\ln s$ or $\ln \widetilde{N}_d$ with all meteorological predictors held constant. The goodness of fit of the regression model is determined by computing the fraction of $R'$ variance that it explains in each grid box, then spatially averaging the results over the domain. On average, the regression method explains 46% of the $R'$ variance when $\ln s$ is the final predictor and 49% of the variance when $\ln \widetilde{N}_d$ is the final predictor. We also apply the method to $R'_{r_e}$, $R'_{\text{LWP}}$, and $R'_{\text{CF}}$ to estimate the Twomey effect, LWP adjustment, and cloud-fraction adjustment.

Our analysis differs from existing global estimates of aerosol indirect effects in several ways. First, estimates that control for fewer meteorological factors are susceptible to bias from correlations between meteorology and aerosols (Mauger and Norris, 2007; Gryspeerdt et al., 2016; Andersen et al., 2017). Our method minimizes this bias by controlling for all of the standard large-scale meteorological drivers of liquid boundary-layer clouds that have been proposed in the literature. Second, estimates derived from daily or monthly grid-box averages of $r_e$ and LWP suffer from aggregation bias because these properties are nonlinearly related to cloud albedo (Feingold et al., 2022; Gryspeerdt et al., 2019). Our method avoids this bias by inferring cloud radiative effects from joint histograms of $r_e$ and LWP rather than grid-box averages. Third, studies of ship tracks, industrial plumes, or volcanic eruptions offer some of the most captivating evidence of aerosol indirect effects, but the estimates they provide may not be representative of the global scale (Possner et al., 2018; Toll et al., 2019; Glassmeier et al., 2021). Our method avoids this potential sampling bias by estimating aerosol indirect effects across the global ocean. Fourth, no global observational study has simultaneously estimated the Twomey effect, LWP adjustment, and cloud-fraction adjustment, so comparisons of these components have been complicated by the fact that each one is estimated using different data, methods, assumptions, and uncertainty quantification (Forster et al., 2021). Our method avoids this complication by estimating all components with a single, self-consistent framework. Our estimates of aerosol indirect effects could still be affected by satellite retrieval biases, but they improve upon existing estimates in these four ways (Painemal and Zuidema, 2011; Ma et al., 2018).

## 3 Global Analysis of Aerosol Indirect Effects

We next present estimates of the Twomey effect and cloud adjustments across the global ocean. The Twomey effect can occur whenever the CCN concentration is small enough that it limits the number of droplets that form in cloud updrafts. This condition is usually satisfied over ocean, so the Twomey effect is expected to be ubiquitous in oceanic clouds (Rosenfeld et al., 2014). Indeed, we find that increasing sulfate concentration is associated with significant cloud brightening from $R_{r_e}$ changes across most of the global ocean (Fig. 2a). Cloud brightening is also observed in response to increasing $\widetilde{N}_d$ (Fig. 2b), but we caution against overinterpreting statistical significance of this relationship because $R_{r_e}$ and $\widetilde{N}_d$ are both inferred from the MODIS $r_e$ retrievals ($\widetilde{N}_d$ is inferred using 10% of the data). Nevertheless, these results show that most marine liquid clouds exhibit the Twomey effect.

In contrast, previous work has shown that cloud adjustments can differ from one cloud regime to another (Zhang et al., 2022). As cloud droplets become smaller, they sediment more slowly out of the cloud-top entrainment zone, and they evaporate more quickly when exposed to entrained air. This enhances evaporation and reduces LWP in non-precipitating clouds (Bretherton et al., 2007; Small et al., 2009). Clouds with smaller droplets also form precipitation more slowly through collision and coalescence. This may cause other changes in cloud properties, including deeper cumulus clouds, longer cloud lifetimes, larger stratiform areas detrained from precipitating cloud elements, or changes in mesoscale cellular structure (Albrecht, 1989; Pincus and Baker, 1994; Rosenfeld et al., 2006; Seifert et al., 2015; Possner et al., 2018; Dagan et al., 2017; Goren et al., 2022). The cloud adjustments can, in turn, affect CCN concentration by changing precipitation scavenging or sulfate formation in cloud droplets (Wood et al., 2012; Kang et al., 2022; Andreae and Rosenfeld, 2008). Some of these mechanisms depend on the meteorological conditions, so they may vary regionally (Chen et al., 2014; Possner et al., 2020; Zhou et al., 2021; Zhang and Feingold, 2023).

The estimated cloud adjustments exhibit regional variations that are consistent with some of these proposed mechanisms. The radiative adjustment from LWP changes is positive in much of the subtropics, and it maximizes in areas of semi-permanent stratocumulus clouds and directly downwind (Fig. 2c-d). This spatial pattern suggests that enhanced evaporation in non-precipitating stratocumulus may contribute to the LWP adjustment (Bender et al., 2018). Weak or insignificant LWP adjustments are found across much of the midlatitude oceans, despite the fact that precipitating clouds occur less frequently in these regions as sulfate aerosols become more abundant (W22). This weak overall LWP adjustment might be partly explained by the fact that different cloud regimes in extratropical cyclones exhibit adjustments that may counteract one another (Naud et al., 2017; McCoy et al., 2018). Furthermore, the radiative adjustment from cloud-fraction changes is negative in most of the subtropics and midlatitudes, suggesting that adjustments in these regions may involve changes in cloud lifetime, size, or morphology as well (Fig. 2e-f). Subtropical stratocumulus regions exhibit offsetting adjustments from LWP changes and cloud-fraction changes. In these cases, as CCN become more abundant, the overall liquid-cloud fraction increases, but the increase is disproportionately large in cloud elements with below-average LWP. This combination is consistent with larger stratiform areas detrained from precipitating clouds or a shift from open to closed

mesoscale cellular convection (Possner et al., 2018; Rosenfeld et al., 2006). Thus, aerosol-driven changes in evaporation and precipitation may both contribute to cloud adjustments. The spatial patterns of the adjustments predicted with $\ln s$ resemble those predicted with $\ln \widetilde{N}_d$, suggesting that the estimated adjustments are robust.

We determine the relative importance of the Twomey effect and cloud adjustments at the global scale by averaging the regression coefficients over ocean. These averages can be interpreted as the cloud radiative anomalies that would occur if sulfate concentration and $\widetilde{N}_d$ were increased by a factor of 2.7 at every location. Uncertainty quantification for these estimates is described in Appendix C. For the regressions against $\ln s$ and $\ln \widetilde{N}_d$, we find that the Twomey effect and cloud-fraction adjustment both significantly increase SW reflection to space (Fig. 3). The cloud-fraction adjustment enhances SW reflection by between 43% and 250% compared to the Twomey effect alone (95% CI), so it makes a substantial contribution to the overall aerosol indirect effect. The large relative magnitude of the cloud-fraction adjustment is consistent with the observed cloud response during a volcanic eruption in Holuhraun, Iceland, and the uncertainty range for this adjustment overlaps with other observational estimates (Chen et al., 2022; Gryspeerdt et al., 2020). Furthermore, the LWP adjustment reduces SW reflection to space, offsetting between 6% and 87% of the Twomey effect. This uncertainty range is comparable to the range of estimates from Diamond et al. (2020) and Gryspeerdt et al. (2019). However, it differs from a ship-track analysis by Manshausen et al. (2022), who find that the LWP adjustment increases SW reflection rather than reduces it. This apparent discrepancy may be a consequence of the fact that localized, short-term aerosol perturbations such as shipping emissions can give rise to different cloud adjustments than sustained aerosol perturbations over larger spatial and temporal scales (Glassmeier et al. 2021). Although the LWP adjustment and cloud-fraction adjustment counteract one another, the total cloud adjustment is still significantly negative, and it is comparable to the Twomey effect. Furthermore, the estimated Twomey effect and total cloud adjustment are similar for the MODIS_CLD and MODIS_CLD+PCL cases, indicating that they do not change substantially when partly cloudy pixels are filtered in different ways. The Twomey effect and total cloud adjustment are also qualitatively consistent when different $N_d$ datasets are used, and they are an order of magnitude larger than $\partial R_{\text{res}}/\partial \ln s$ and $\partial R_{\text{res}}/\partial \ln \widetilde{N}_d$ (Supplementary Fig. 1 and 2). These results robustly show that the Twomey effect and total cloud adjustment cause comparable changes in top-of-atmosphere SW flux.

A limitation of these results is that the MODIS_CLD and MODIS_CLD+PCL cases have offsetting differences in the estimated LWP and cloud-fraction adjustments (Fig. 3). This means that estimates of the individual LWP and cloud-fraction adjustments depend on filtering of partly cloudy pixels, but the estimate of the total cloud adjustment does not. One implication is that aerosol variations must be associated with changes in the relative amounts of partly and fully cloud-covered pixels. Partly and fully cloudy pixels reside on cloud edges and interiors, respectively, so a change in the relative amounts of these pixels implies a change in the cloud perimeter-to-area ratio (W22). This suggests that the global cloud adjustment may involve changes in cloud size or morphology. A case study demonstrating this concept is presented in Appendix D. A second implication is that the conventional practice of estimating the individual LWP and cloud-fraction adjustments at the global scale will inevitably lead to results that depend on the classification and filtering of partly cloudy pixels. Thus, instrument sensitivity, horizontal resolution, and subjective pixel-classification thresholds can all affect the

results. In contrast, we find that more robust results can be obtained by estimating the total cloud adjustment. Our analysis provides the first direct assessment of this quantity from observations, reanalysis, and radiative transfer modeling.

## 4 Implications for Historical Aerosol Forcing

We next combine estimates of aerosol indirect effects and historical sulfate changes to infer $ERF_{aci}$. The forcing estimates use sulfate rather than $\widetilde{N}_d$ because sulfate concentration is a widely available output variable from GCMs, but $\widetilde{N}_d$ is not. Assuming that sulfate dominates the anthropogenic influence on CCN (Charlson et al., 1992; Stevens 2015), we estimate SW $ERF_{aci}$ from liquid-topped clouds according to

$$ERF_{aci} \approx \frac{\partial R}{\partial \ln s} \Delta \ln s$$

where $\Delta \ln s$ is the change in sulfate concentration between preindustrial (1850-1859) and present-day (2005-2014) conditions simulated by GCMs that participated in CMIP6. This method of estimating $ERF_{aci}$ has been validated with volcanic eruptions and other known variations of regional sulfur-dioxide emissions (W22).

We note that this method differs from that of a similar study by W22 in three ways. First, we estimate $ERF_{aci}$ from all liquid-topped clouds, while W22 estimate $ERF_{aci}$ from low-level clouds, defined as clouds with tops between the surface and 680 hPa. We applied their method to estimate SW $ERF_{aci}$ from all liquid-topped clouds, and we found that the result is about 26% larger in magnitude than the estimate of SW $ERF_{aci}$ from low-level clouds. Second, we estimate SW $ERF_{aci}$, while W22 estimate net $ERF_{aci}$. Thus, their estimate includes an additional $ERF_{aci}$ component from changes in longwave radiation, which offsets about 14% of the SW component (Appendix B). Third, we estimate $ERF_{aci}$ with MODIS observations and radiative kernels, while W22 estimate $ERF_{aci}$ with CERES observations for their main result. Our kernel-based estimates of $R'$ are about 4.6% larger in magnitude than the corresponding CERES observations (Appendix A). These three factors cause our $ERF_{aci}$ estimates to have a larger magnitude than those reported by W22.

Our method for estimating $ERF_{aci}$ can be applied to each component of the aerosol indirect effect to quantify the associated historical effective radiative forcing. The Twomey effect contributes a negative instantaneous radiative forcing ($IRF_{aci}$) that peaks in subtropical stratocumulus regions and the midlatitude oceans of the Northern Hemisphere (Fig. 4a). Forcing is relatively large in stratocumulus regions because these clouds exhibit a strong radiative response to CCN perturbations (Fig. 2), and forcing is relatively large in the Northern Hemisphere midlatitudes because these regions are close to anthropogenic aerosol sources. The geographic pattern and magnitude of the $IRF_{aci}$ generally agree with the estimates of McCoy et al. (2017), Kinne (2019), and Jia et al. (2021). Furthermore, the $IRF_{aci}$ is comparable to the effective radiative forcing from the combined effect of LWP adjustments ($A_{LWP}$) and cloud-fraction adjustments ($A_{CF}$) (Fig. 4b). Thus, the magnitude of the overall $ERF_{aci}$ peaks in the subtropical stratocumulus regions and the Northern Hemisphere midlatitudes as well (Fig. 4c).

We average the forcing components over ocean between 55°S and 55°N to quantify their large-scale climate impacts. Confidence intervals are computed accounting for regression-slope uncertainty and inter-model spread in the estimates of $\Delta \ln s$ (Appendix C). To frame our results in the context of the existing literature, we compare our estimates with forcing calculations from an assessment of the World Climate Research Programme (WCRP) (Bellouin et al., 2020) and forcing estimates from 14 GCMs that participated in the Coupled Model Intercomparison Project Phase 5 (CMIP5) and AeroCom experiments computed by Gryspeerdt et al. (2020) (Supplementary Table 1). We repeat the original WCRP analysis, except that we restrict the calculation to SW forcing over ocean between 55°S and 55°N, as described in the Supplementary Information. The GCM forcing estimates are averaged over ocean between 55°S and 55°N as well. Although all forcing estimates are computed over the same spatial domain, their time periods differ slightly from one another: The present-day reference years are 2005-2014 for our estimates, 2005-2015 for the WCRP estimates, and 2000 for the CMIP5 and AeroCom estimates, and the preindustrial reference years are 1850-1859 for our estimates, 1850 for the WCRP and CMIP5 estimates, and 1860 for the AeroCom estimates (Bellouin et al., 2020; Zelinka et al., 2014; Ghan et al., 2016). These differences in the reference periods could cause differences in ERF$_{aci}$ of 0.1 W m$^{-2}$ or less (IPCC, 2021).

Averages of IRF$_{aci}$, cloud adjustments, and the overall ERF$_{aci}$ are compared in Fig. 5. We find that A$_{CF}$ is significantly negative, and A$_{LWP}$ is more likely than not to be positive, but the magnitudes of these estimates depend on filtering choices for partly cloudy pixels. The three remaining components are insensitive to partly-cloudy-pixel filtering: the IRF$_{aci}$ is $-0.77 \pm 0.25$ W m$^{-2}$, the total cloud adjustment is $-1.02 \pm 0.43$ W m$^{-2}$, and the overall ERF$_{aci}$ is $-1.86 \pm 0.62$ W m$^{-2}$ (95% CIs from MODIS$_{CLD}$). These results lie inside the ranges of the corresponding WCRP and GCM estimates, and the median IRF$_{aci}$ agrees very well with the WCRP and GCM values. However, our results reduce uncertainty of each component by at least 62% relative to the confidence intervals of the WCRP and at least 23% relative to the range of GCMs. Furthermore, the WCRP and GCM estimates do not rule out the possibility that the total cloud adjustment is positive or an order of magnitude smaller than the IRF$_{aci}$. According to our analysis, however, such a small or positive adjustment is implausible. Thus, our findings reduce uncertainty in the historical IRF$_{aci}$ and total cloud adjustment, and they clarify the relative importance of these components.

The ocean-average SW ERF$_{aci}$ can be scaled to establish an upper bound for the global-average net ERF$_{aci}$. Let ERF$_{aci,net,g}$ be the global-average net ERF$_{aci}$, and let ERF$_{aci,net,d}$ be the domain-average net ERF$_{aci}$, where the domain includes ocean areas between 55°S and 55°N. Assuming that the average net ERF$_{aci}$ is negative outside the domain (Diamond et al., 2020), it follows that

$$\text{ERF}_{aci,net,g} < f\,\text{ERF}_{aci,net,d} \tag{1}$$

where $f = 0.56$ is the fraction of global surface area covered by the domain. ERF$_{aci,net,d}$ can be expressed as

$$\text{ERF}_{aci,net,d} = \text{ERF}_{aci,sw,d}(1 - \beta) \tag{2}$$

where ERF$_{aci,sw,d}$ is the domain-average SW ERF$_{aci}$ and $\beta$ is the fraction of SW ERF$_{aci}$ that is offset by longwave ERF$_{aci}$. Our radiative kernel cannot accurately assess longwave cloud radiative effects, so $\beta$ is estimated by applying cloud-controlling

factor analysis to CERES satellite data instead (Appendix B). The resulting uncertainty range is $\beta = 0.14 \pm 0.06$ (90% CI). We evaluate equation (2) with the bounds of the 90% CIs of $\beta$ and $ERF_{aci,sw,d}$ from the MODIS$_{CLD}$ and MODIS$_{CLD+PCL}$ cases, then select the least-negative value of $ERF_{aci,net,d}$ to evaluate inequality (1). The result constitutes an upper bound for $ERF_{aci,net,g}$.

The above reasoning implies a 95% probability that the global net $ERF_{aci}$ from liquid clouds is more negative than $-0.56$ W m$^{-2}$ (relative to 1850-1859). Equivalent upper bounds from the published literature include $-0.07$ W m$^{-2}$ from the WCRP assessment and $-0.3$ W m$^{-2}$ from the observation-based estimate of the IPCC Sixth Assessment Report (relative to 1850 and 1750, respectively) (Bellouin et al., 2020; Forster et al., 2021). Our analysis thus supports a more stringent upper bound on global $ERF_{aci}$. This constraint is similar to another estimate from cloud-controlling factor analysis presented by

W22, but our estimate invokes weaker assumptions because it does not extrapolate forcing to areas outside the domain. Our upper-bound estimate also complements evidence from global energy-balance arguments, which constrains the lower bound of $ERF_{aci}$ (Stevens et al., 2015; Smith et al., 2021).

## 5 Conclusion

    We analyze MODIS satellite data and adapt techniques from the cloud-feedback literature to quantify aerosol

indirect effects from liquid-topped clouds. Our method avoids aggregation and sampling biases that may affect some previous studies, and it controls for all of the standard large-scale meteorological drivers of liquid boundary-layer clouds that have been proposed in the literature, thereby minimizing biases from confounding meteorological factors (Possner et al., 2018; Glassmeier et al., 2021; Feingold et al., 2022). Furthermore, the Twomey effect, LWP adjustment, and cloud-fraction adjustment are simultaneously quantified with a single, self-consistent framework. This guarantees that all of the

components, and their uncertainties, are quantified in a consistent way. Although it is important to continue characterizing satellite retrieval biases, to include new meteorological cloud-controlling factors as they are discovered, to investigate non-linear and non-local relationships between clouds and their controlling factors (Lewis et al., 2023), and to quantify additional $ERF_{aci}$ components from ice-containing clouds, our method overcomes several limitations that affect previous observational estimates of aerosol indirect effects.

We apply our method to constrain aerosol indirect effects across the global ocean. We find that increasing CCN concentration is associated with widespread cloud brightening from the Twomey effect, a positive radiative adjustment from decreasing LWP in subtropical stratocumulus regions, and a negative radiative adjustment from increasing cloud fraction in the subtropics and midlatitudes. The estimated aerosol indirect effects are combined with historical sulfate changes from CMIP6 models to quantify the associated SW $ERF_{aci}$. The Twomey effect and total cloud adjustment are estimated to

contribute $-0.77 \pm 0.25$ W m$^{-2}$ and $-1.02 \pm 0.43$ W m$^{-2}$, respectively, to the SW $ERF_{aci}$ averaged over ocean between 55°S and 55°N (95% CIs). Our findings reduce uncertainty in these components of aerosol forcing and suggest that liquid-cloud adjustments make a larger contribution to the forcing than is commonly believed.

## Appendix A: Cloud Radiative Kernel

We compute a SW cloud radiative kernel for the MODIS $r_e$-LWP joint histogram to quantify the effect of liquid-cloud anomalies on the top-of-atmosphere SW flux. The radiative kernel is similar to that of Zelinka et al. (2012) with two exceptions. First, our kernel represents liquid-topped clouds, while their kernel represents clouds of all phases. Second, our kernel is partitioned by $r_e$ and LWP, while their kernel is partitioned by cloud-top pressure and cloud optical thickness. Besides these exceptions, we calculate the kernel by closely following the method of Zelinka et al. (2012).

The first step of the kernel calculation is to quantify the clear-sky upward SW flux at the top of the atmosphere for various combinations of surface albedo, latitude, and calendar month. Calculations are performed with the Rapid and Accurate Radiative Transfer Model for Global Climate Models (Clough et al., 2005) using inputs that include the climatological seasonal cycle of humidity from MERRA-2, a standard ozone profile, and a solar constant of 1361 W m$^{-2}$. For a given latitude and month, we chose a day in the middle of the month and calculate the average of the cosine of the solar zenith angle $\mu_i$ for each one-hour interval throughout the day. We then scale $\mu_i$ by $SW_{\downarrow,\mathrm{CERES}}/SW_{\downarrow,\mathrm{day}}$, where $SW_{\downarrow,\mathrm{day}}$ is the daily-mean insolation for the day in the middle of the month and $SW_{\downarrow,\mathrm{CERES}}$ is the monthly-mean insolation from CERES satellite data (Loeb et al., 2018). This step ensures that the monthly-mean insolation for the radiative kernel is equal to that of CERES. We compute the clear-sky SW flux for each of the 24 $\mu_i$ terms, then average the results. The calculations are performed using surface albedo of 0, 0.5, and 1. The final result is a matrix of clear-sky upward SW flux at the top of the atmosphere as a function of surface albedo, latitude, and calendar month.

The next step is similar to the clear-sky calculations except that an overcast and horizontally uniform liquid cloud is introduced in the radiation code. The $r_e$ and LWP of the cloud are varied, and cloud-top pressure is set to 850 hPa to match the modal value retrieved by MODIS. For $r_e$, we use the standard MODIS retrieval algorithm, $r_{e,\mathrm{std}}$, and the 3.7 $\mu$m retrieval algorithm, $r_{e,3.7}$. Monthly gridded values of $r_{e,\mathrm{std}}$ and $r_{e,3.7}$ have a correlation coefficient of 0.92 over ocean, but they are generally different because $r_{e,3.7}$ represents conditions closer to the cloud top (Platnick, 2000). We therefore prescribe $r_e$ inside the cloud as

$$r_e = \begin{cases} r_{e,3.7}, & \tau_c < 3 \\ \tilde{r}_{e,\mathrm{std}}, & \tau_c \geq 3 \end{cases}$$

where $\tau_c$ is the visible optical depth below cloud top and $\tilde{r}_{e,\mathrm{std}} \equiv m r_{e,3.7} + b$, where $m = 1.14$ and $b = -0.35$ $\mu$m. The coefficients $m$ and $b$ are determined by regressing $r_{e,\mathrm{std}}$ against $r_{e,3.7}$ using all monthly $1° \times 1°$ ocean grid boxes, and the $\tau_c = 3$ threshold is chosen from weighting functions estimated by Platnick (2000). Using these relationships, we calculate the top-of-atmosphere SW flux with different combinations of $r_{e,3.7}$ and LWP that correspond to the bins of the MODIS $r_e$-LWP joint histogram. Separate calculations are performed for synthetic clouds at the four edges of each bin, and the results are averaged to get one value of upward SW flux at the top of the atmosphere for each bin. We then subtract the resulting value from the clear-sky upward SW flux to determine the SW cloud radiative effect (CRE). These calculations produce a matrix of SW CRE above an overcast liquid cloud as a function of latitude, surface albedo, calendar month, $r_e$, and LWP.

The final step of the calculation is to convert the overcast-sky SW CRE to a cloud radiative kernel, $K$. Let $C_{rl}$ be the liquid-cloud fraction in effective radius bin $r$ and LWP bin $l$. The $K$ matrix represents how anomalies of $C_{rl}$ affect $R$ with all non-cloud factors fixed:

$$K_{rl} \equiv \frac{\partial R}{\partial C_{rl}}.$$

$K_{rl}$ is computed by dividing the overcast-sky SW CRE by 100%. We apply linear interpolation to transform $K_{rl}$ from latitude-surface-albedo space to latitude-longitude space using the climatological seasonal cycle of clear-sky surface albedo from CERES. The final radiative kernel has units of W m$^{-2}$ %$^{-1}$ and is a function of latitude, longitude, calendar month, $r_e$, and LWP (Fig. 1b).

We validate the radiative kernel by comparing $R'$ observations from the CERES FluxByCldTyp dataset (Sun et al., 2022) with $R'$ estimates computed from the kernel and MODIS $r_e$-LWP joint histograms. The kernel estimates are regressed against the CERES observations using data from all monthly 1° × 1° ocean gridboxes between 55°S and 55°N from 2003 through 2020. The regression slope is $1.046 \pm 0.005$ (95% CI) (Fig. A1). Thus, biases of our radiative kernel and differences between the MODIS and CERES cloud retrieval algorithms cause the kernel-based values of $R'$ to overestimate the magnitude of their CERES counterpart by $+4.6 \pm 0.5\%$.

**Appendix B: Decomposing Cloud Radiative Effects**

For a given latitude, longitude, and time, the total liquid-cloud-induced SW flux anomaly at the top of the atmosphere, $R'$, is

$$R' = \sum_{r=1}^{6} \sum_{l=1}^{7} K_{rl} C_{rl}' \tag{B1}$$

We decompose the term on the right side of equation B1 to estimate how much cloud-amount anomalies, $r_e$ anomalies, and LWP anomalies contribute to $R'$. The decomposition closely follows the method described in Appendix B of Zelinka et al. (2013), except that our radiative kernel and histogram have different dimensions.

First, let $C_{\text{tot}}$ be the total liquid-cloud fraction summed over all histogram bins. We express $C_{rl}'$ as

$$C_{rl}' = \frac{\bar{C}_{rl}}{\bar{C}_{\text{tot}}} C_{\text{tot}}' + C_{rl}^* \tag{B2}$$

where overbars denote values from the local climatological seasonal cycle. The first term on the right side of equation B2 represents the anomalies of $C_{rl}$ that would occur if $C_{\text{tot}}'$ were distributed among the $r_e$-LWP bins such that the normalized distribution in the histogram remains the same as the climatology. In other words, this term accounts for a change in total liquid-cloud fraction, holding fixed the proportion of clouds in each histogram bin. The second term on the right side of equation B2 accounts for anomalies of $C_{rl}$ that remain after removing $(\bar{C}_{rl}/\bar{C}_{\text{tot}})C_{\text{tot}}'$. This term represents shifts in the

distribution of $r_e$ and LWP with the total liquid-cloud fraction fixed. By construction, $C_{rl}^*$ vanishes when it is summed over all histogram bins.

Next, we decompose the radiative kernel into two terms:

$$K_{rl} = K_0 + \widehat{K}_{rl}. \tag{B3}$$

Here, $K_0$ is the average of $K_{rl}$ weighted by the climatological cloud fraction,

$$K_0 \equiv \sum_{r=1}^{6} \sum_{l=1}^{7} \frac{\bar{C}_{rl}}{\bar{C}_{\text{tot}}} K_{rl}, \tag{B4}$$

and $\widehat{K}_{rl} \equiv K_{rl} - K_0$. With the relationships in equations B1-B4, $R'$ can be expressed as

$$R' = K_0 C'_{\text{tot}} + \sum_{r=1}^{6} \sum_{l=1}^{7} \widehat{K}_{rl} C_{rl}^*.$$

We next decompose $\widehat{K}_{rl}$ into three components:

$$\widehat{K}_{rl} = \widehat{K}_r + \widehat{K}_l + \widehat{K}_{\text{res}}$$

where

$$\widehat{K}_r \equiv \sum_{l=1}^{7} \left( \widehat{K}_{rl} \sum_{r=1}^{6} \frac{\bar{C}_{rl}}{\bar{C}_{\text{tot}}} \right),$$

$$\widehat{K}_l \equiv \sum_{r=1}^{6} \left( \widehat{K}_{rl} \sum_{l=1}^{7} \frac{\bar{C}_{rl}}{\bar{C}_{\text{tot}}} \right),$$

and

$$\widehat{K}_{\text{res}} \equiv \widehat{K}_{rl} - \widehat{K}_r - \widehat{K}_l.$$

$R'$ can then be expressed as

$$R' = K_0 C'_{\text{tot}} + \sum_{r=1}^{6} \left( \widehat{K}_r \sum_{l=1}^{7} C_{rl}^* \right) + \sum_{l=1}^{7} \left( \widehat{K}_l \sum_{r=1}^{6} C_{rl}^* \right) + \sum_{r=1}^{6} \sum_{l=1}^{7} \widehat{K}_{\text{res}} C_{rl}^*. \tag{B5}$$

The first term on the right side of equation B5 is the SW flux anomaly that would occur if the anomaly of total liquid-cloud fraction were distributed among the $r_e$-LWP bins such that the proportion of cloud fraction in each bin is the same as the climatology. This term represents the contribution of cloud-amount anomalies to $R'$. The second term on the right side results from multiplying an effective kernel that accounts for systematic variations in $r_e$ by the change in cloud fraction at each $r_e$

bin. This term represents the contribution of $r_e$ anomalies to $R'$ with LWP and total liquid-cloud fraction held fixed. The third term on the right side is similar to the second term except that it represents the contribution of LWP anomalies to $R'$ with $r_e$ and total liquid-cloud fraction held fixed. The final term on the right is the residual of the decomposition. The cloud-amount, $r_e$, LWP, and residual components of $R'$ are denoted by $R'_{\text{CF}}$, $R'_{r_e}$, $R'_{\text{LWP}}$, and $R'_{\text{res}}$, respectively. In the Supplementary Information, we validate the $R'$ decomposition using synthetic-data test cases in which $R'_{r_e}$ and $R'_{\text{LWP}}$ can be

estimated theoretically with the two-stream radiative transfer approximation (Supplementary Fig. 3). We also verify that $R'_{r_e}$ and $R'_{\text{LWP}}$ are similar when different common assumptions are made about cloud vertical structure, including a vertically uniform cloud model, an adiabatic cloud model, and the two-layer cloud model from the kernel calculation.

The final step of the decomposition is to adjust $R'_{\text{CF}}$ to account for obscuration effects from non-liquid clouds. Because MODIS is a passive instrument, changes in non-liquid clouds can artificially change the retrieved liquid-cloud 400 fraction if they obscure liquid clouds from the satellite view. We control for these obscuration effects by replacing $C'_{\text{tot}}$ in equation B5 by $[C_{\text{tot}}/(100\% - I_{\text{tot}})]'(100\% - \bar{I}_{\text{tot}})$, where $I_{\text{tot}}$ is the retrieved fraction of non-liquid clouds. This change of variables is adapted from the procedure recommended by Scott et al. (2020).

We also wish to compare the SW and longwave (LW) components of cloud radiative effects. However, our radiative kernel assumes a constant cloud-top pressure, so it cannot accurately assess the LW component (Appendix A). 405 Instead, we analyze observations of the SW and LW radiative effects of liquid-topped clouds from the CERES FluxByCldTyp dataset. We perform cloud-controlling factor analysis with the CERES data to attain estimates of ERF$_{\text{aci}}$ in addition to those presented in the main paper. The CERES-based estimate of ERF$_{\text{aci}}$ averaged over ocean between 55°S and 55°N is $-1.52 \pm 0.51$ W m$^{-2}$ for SW radiation and $+0.21 \pm 0.13$ W m$^{-2}$ for LW radiation. The LW component offsets $14 \pm 7\%$ of the SW component (95% CIs). These values determine $\beta$ in the upper-bound estimate of global net ERF$_{\text{aci}}$ (equation 410 2).

**Appendix C: Uncertainty**

Uncertainty in our ocean-average ERF$_{\text{aci}}$ estimates arises from uncertainty in the regression coefficients representing $\partial R/\partial \ln s$ and uncertainty in the model estimates of $\Delta \ln s$. We first quantify the component that is attributable to regression-coefficient uncertainty. For a particular grid box $i$, let $\epsilon_i$ represent the half-width of the 95% confidence 415 interval of grid-box mean ERF$_{\text{aci}}$. We estimate $\epsilon_i$ as

$$\epsilon_i = t_i \sigma_i \sqrt{\frac{N_{\text{nom},i}}{N_{\text{eff},i}}} [\Delta \ln s]_i$$

where $\sigma_i$ is the standard error of the regression coefficient, $N_{\text{nom},i}$ is the nominal number of temporal degrees of freedom (i.e. the number of months in the record), $N_{\text{eff},i}$ is the effective number of temporal degrees of freedom, square brackets indicate the central estimate of a parameter, and $t_i$ is the critical value of a Student's $t$-distribution at the $(1 - \alpha/2)100\%$ 420 significance level using $N_{\text{eff},i} - 8$ degrees of freedom and $\alpha = 0.05$. The ratio $N_{\text{nom},i}/N_{\text{eff},i}$ is estimated as $(1 + a)/(1 - a)$, where $a$ is the temporal lag-1 autocorrelation of $R'_i$. The $\epsilon_i$ terms are then combined to account for spatial averaging over the domain. Uncertainty of the domain-average forcing, $\delta_{\text{obs}}$, is

$$\delta_{\text{obs}} = \frac{\sqrt{\sum_{i=1}^{N_{\text{nom}}^*} w_i^2 \epsilon_i^2}}{\sum_{i=1}^{N_{\text{nom}}^*} w_i} \sqrt{\frac{N_{\text{nom}}^*}{N_{\text{eff}}^*}}$$

where $N_{\text{nom}}^*$ is the nominal number of spatial degrees of freedom (i.e. the number of latitude-longitude grid-boxes in the domain), $N_{\text{eff}}^*$ is the effective number of spatial degrees of freedom, and $w_i$ is the ocean area in grid box $i$. The ratio $N_{\text{nom}}^*/N_{\text{eff}}^*$ is estimated by applying equation 5 of Bretherton et al. (1999) to the gridded $R'$ data. The resulting value of $\delta_{\text{obs}}$ represents the half-width of the 95% confidence interval of $ERF_{\text{aci}}$ that is attributable to regression-coefficient uncertainty. Confidence intervals for the spatial averages of $\partial R / \partial \ln s$ and $\partial R / \partial \ln \widetilde{N}_d$ are calculated similarly.

The second source of uncertainty of $ERF_{\text{aci}}$ arises from inter-model spread in the estimates of $\Delta \ln s$. Because we have estimates from 20 climate models, we construct a 95% confidence interval for $ERF_{\text{aci}}$ that excludes one model and encompasses the range of the other 19. We first calculate 20 estimates of $ERF_{\text{aci}}$ by multiplying $\Delta \ln s$ from each model by $[\partial R / \partial \ln s]$. The half-width of the confidence interval, $\delta_{\ln s}$, is estimated as the minimum of $|c_1 - c_{19}|/2$ and $|c_2 - c_{20}|/2$, where $c_1$, $c_2$, $c_{19}$, and $c_{20}$ are the smallest, second smallest, second largest, and largest values of the 20 $ERF_{\text{aci}}$ estimates, respectively. This uncertainty analysis accounts for inter-model differences in aerosol processing, but it does not account for uncertainty in anthropogenic sulfur-dioxide emissions because the climate-model simulations apply the same emission values. However, sulfur-dioxide emissions depend primarily on the sulfur content of fuel rather than the conditions of combustion, so global emission inventories have a relatively small uncertainty of about $\pm 11\%$ (90% CI from Smith et al., 2011). This is expected to cause an equivalent fractional uncertainty in the global burden of anthropogenic sulfate aerosol (Charlson et al., 1992; Stevens, 2015). In contrast, inter-model differences in aerosol processing lead to an uncertainty of $\pm 43\%$ in the change in $s$ since 1850 averaged over ocean (90% CI estimated by computing the interval that includes 18 of the 20 CMIP6 models). The quadrature sum of these two components determines their combined uncertainty, so uncertainty in aerosol processing dominates the overall uncertainty in global-mean $\Delta \ln s$. Approximating $\delta_{\ln s}$ from inter-model differences is therefore justified. Finally, we note that $ERF_{\text{aci}}$ strongly depends on the preindustrial aerosol state, so our estimates are approximately valid as long as the inter-model spread in preindustrial CCN concentration encompasses the true values (Carslaw et al., 2013; Kinne, 2019). The possibility that CMIP6 models have systematic biases that violate this condition cannot be ruled out at this time.

The estimated $\delta_{\text{obs}}$ and $\delta_{\ln s}$ represent independent sources of uncertainty, so they are combined in quadrature. The overall 95% confidence interval is given by $ERF_{\text{aci}} \pm \sqrt{\delta_{\text{obs}}^2 + \delta_{\ln s}^2}$. Confidence intervals for $IRF_{\text{aci}}$, $A_{\text{LWP}}$, $A_{\text{CF}}$, and the total cloud adjustment are calculated similarly.

## Appendix D: Filtering of Partly Cloudy Pixels

Different filtering methods for partly cloudy pixels in the $MODIS_{\text{CLD}}$ and $MODIS_{\text{CLD+PCL}}$ cases lead to offsetting differences in the estimated LWP and cloud-fraction adjustments (Fig. 3). One possible explanation for this discrepancy is

that CCN anomalies cause changes in the morphology or horizontal size of liquid clouds, thereby changing the relative amounts of partly and fully cloudy pixels (Possner et al., 2018; Rosenfeld et al., 2006). Here we examine a case study to demonstrate this concept.

We analyze instantaneous pixel data from the MODIS MOD06_L2 dataset collection 6.1 (Platnick et al., 2015) obtained during a single overpass of the Terra satellite on September 27, 2019. On this day, MODIS measured stratocumulus clouds in the Southeast Pacific Ocean with different forms of mesoscale cellular convection. The clouds in box C in Fig. D1a mostly exhibit closed cells, and the clouds in box O mostly exhibit open cells. We select data from these boxes and bin the liquid-cloud pixels into histograms of cloud fraction partitioned by LWP. Let $C_l$ represent cloud fraction in LWP bin $l$. In one case, $C_l$ and $C_{\text{tot}}$ are computed by counting only the fully cloudy pixels (CLD), and in a second case, $C_l$ and $C_{\text{tot}}$ are computed by counting both fully and partly cloudy pixels (CLD+PCL). Box C contains mostly fully cloudy pixels, so the two cases have similar values of $C_l$ and $C_{\text{tot}}$ (Fig. D1b). In contrast, box O contains broken clouds that have a smaller horizontal scale, a larger perimeter-to-area ratio, and a larger proportion of partly cloudy pixels. The partly cloudy pixels cover about 13% of the area of box O, and their retrieved LWP is usually smaller than that of the fully cloudy pixels in the box. This causes a difference in $C_{\text{tot}}$ and the $C_l$ distribution between the CLD and CLD+PCL cases (Fig. D1c). Filtering of partly cloudy pixels therefore affects the grid-box-level statistics of cloud fraction and LWP in this example.

We next examine differences between the cloud-fraction histograms of the two boxes to demonstrate the implications for estimating $R'$. For the purpose of this demonstration, let the baseline cloud population be defined by the clouds in Box O, and let the cloud-fraction anomalies be defined by the cloud fraction in box C minus the cloud fraction in box O. The baseline and anomalies are indicated by overbars and primes, respectively. $C'_l$ can be decomposed according to

$$C'_l = \frac{\bar{C}_l}{\bar{C}_{\text{tot}}} C'_{\text{tot}} + C^*_l$$

where $C^*_l \equiv C'_l - \frac{\bar{C}_l}{\bar{C}_{\text{tot}}} C'_{\text{tot}}$. This decomposition is equivalent to equation B2 except that it is performed on a one-dimensional LWP histogram rather than a two-dimension LWP-$r_e$ joint histogram. The first term on the right side of the equation determines $R'_{\text{CF}}$, and the second term determines $R'_{\text{LWP}}$. Compared with the CLD case, the CLD+PCL case has a smaller value of $C'_{\text{tot}}$ (Fig. D1d), which reduces the magnitude of $R'_{\text{CF}}$. Furthermore, because the partly cloudy pixels in box O occupy the smallest LWP bins, including these pixels in the histogram causes a less extreme shift in the $C'_l$ distribution towards small LWP values between Box O and Box C. This reduces the magnitude of $R'_{\text{LWP}}$ in the CLD+PCL case relative to that of the CLD case. Thus, including partly cloudy pixels in the histograms leads to offsetting changes in $R'_{\text{CF}}$ and $R'_{\text{LWP}}$ that reduce the magnitude of both terms. Our estimates of global cloud adjustments depend on filtering of partly cloudy pixels in a similar way, suggesting that the adjustments may involve changes in cloud size or morphology as well.

## Code Availability

Matlab code used to analyze data can be obtained by contacting the corresponding author. The Rapid and Accurate Radiative Transfer Model for GCMs is publicly available at http://rtweb.aer.com/rrtm_frame.html.

## Data Availability

All satellite data, reanalysis, and GCM output used in this study are publicly available. MODIS cloud histograms are available at https://ladsweb.modaps.eosdis.nasa.gov/archive/allData/62/, MODIS $N_d$ data are available at https://catalogue.ceda.ac.uk/uuid/864a46cc65054008857ee5bb772a2a2b, CERES data are available at https://ceres.larc.nasa.gov/data/, MERRA-2 data are available at https://disc.gsfc.nasa.gov/, and CMIP6 output is available at https://esgf-node.llnl.gov/projects/cmip6/. The SW cloud radiative kernel is available at: https://github.com/caseywall7926/MODIS_Re-LWP_kernel. Estimates of the components of aerosol indirect effects and ERF$_{aci}$ averaged over ocean are listed in Supplementary Table 3.

## Author Contribution

CJW designed research, performed research, analyzed data, and wrote the original paper draft. TS and AP contributed ideas that shaped the study. All authors helped revise the original paper draft.

## Competing Interests

The authors declare that they have no conflict of interest.

## Acknowledgements

This project has received funding from the European Union's Horizon 2020 research and innovation programme under the Marie Skłodowska-Curie grant agreement No 101019911. We thank Daniel McCoy, Ed Gryspeerdt, Robert Pincus, and Velle Toll for helpful discussions, and we thank Ying Chen and Jianhao Zhang for providing constructive reviews that improved the manuscript.

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

**Figures**

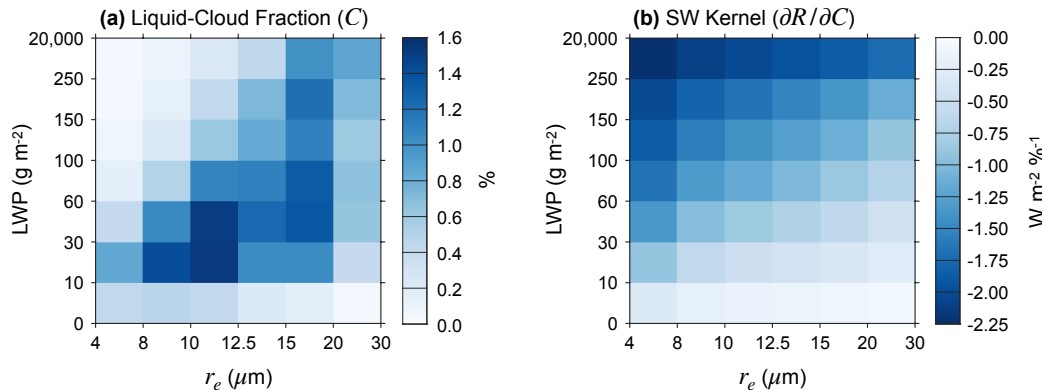

**Figure 1.** MODIS joint histogram and SW cloud radiative kernel averaged over the latitude, longitude, and time dimensions. Averages are computed over ocean between 55°S and 55°N. (a) Joint histogram of liquid-cloud fraction ($C$) partitioned by liquid water path (LWP) and cloud-droplet effective radius ($r_e$). (b) SW cloud radiative kernel. The kernel represents $\partial R/\partial C$, where $R$ is the SW radiative effect of liquid clouds at the top of the atmosphere.

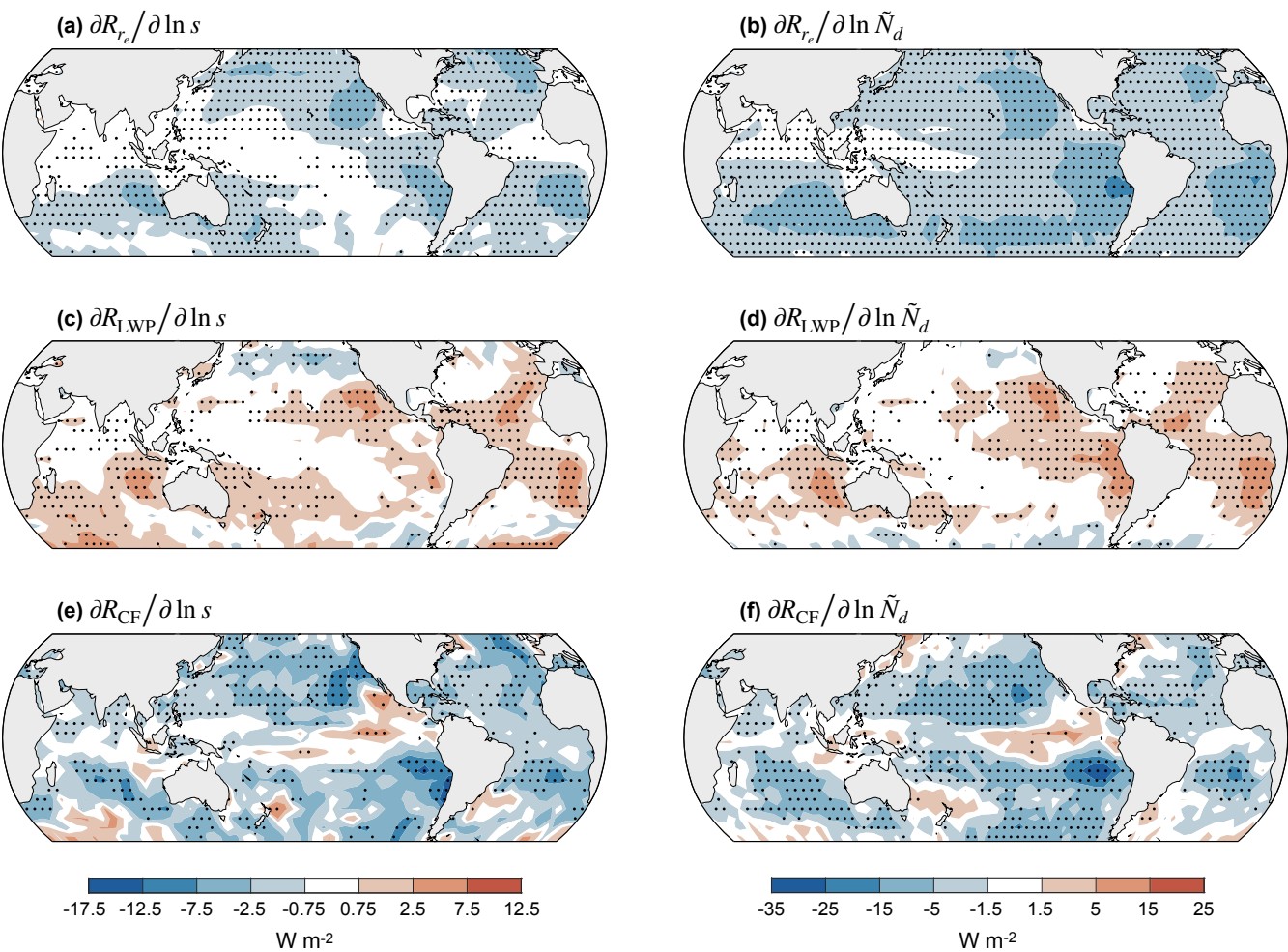

**Figure 2.** Aerosol indirect effects estimated with two indicators of cloud-base CCN concentration: sulfate aerosol mass concentration at 910 hPa ($s$) and cloud-droplet number concentration from pixels with the largest 10% cloud optical thickness ($\widetilde{N}_d$). Linear regression coefficients are shown for (a) $\partial R_{r_e}/\ln s$, (b) $\partial R_{r_e}/\ln \widetilde{N}_d$, (c) $\partial R_{\text{LWP}}/\ln s$, (d) $\partial R_{\text{LWP}}/\ln \widetilde{N}_d$, (e) $\partial R_{\text{CF}}/\ln s$, and (f) $\partial R_{\text{CF}}/\ln \widetilde{N}_d$, where $R_{r_e}$, $R_{\text{LWP}}$, and $R_{\text{CF}}$ are the top-of-atmosphere SW flux perturbations caused by $r_e$ anomalies, LWP anomalies, and cloud-fraction anomalies, respectively. (a-b) represents the Twomey effect, (c-d) represents the LWP adjustment, and (e-f) represents the cloud-fraction adjustment. Stippling indicates regression coefficients that are significantly different from zero with the false discovery rate limited to 0.1 (Wilks, 2016). Cloud radiative effects are computed with only fully cloud-covered pixels included in the cloud histograms (MODIS$_{\text{CLD}}$). Note that the contour values in (a), (c), and (e) are proportional to those in (b), (d), and (f).

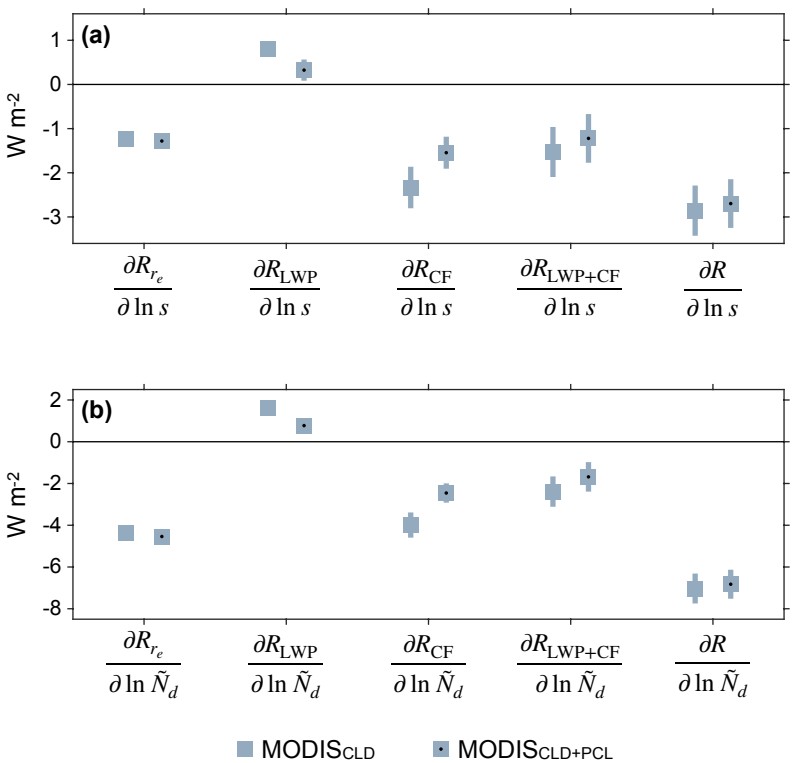

**Figure 3.** Spatial averages of the regression coefficients that represent aerosol indirect effects. Averages are computed over ocean between 55°S and 55°N. (a) Aerosol indirect effects estimated with $s$ as the CCN indicator. $\partial R_{\text{LWP+CF}}/\partial \ln s$ represents the total cloud adjustment ($R'_{\text{LWP+CF}} \equiv R'_{\text{LWP}} + R'_{\text{CF}}$). The MODIS$_{\text{CLD}}$ case is computed with only fully cloud-covered pixels included in the cloud histograms, and the MODIS$_{\text{CLD+PCL}}$ case is computed with both fully and partly cloud-covered pixels included. (b) Similar to (a), except that $\tilde{N}_d$ is the CCN indicator. Squares show mean values, and vertical lines show 95% CIs.

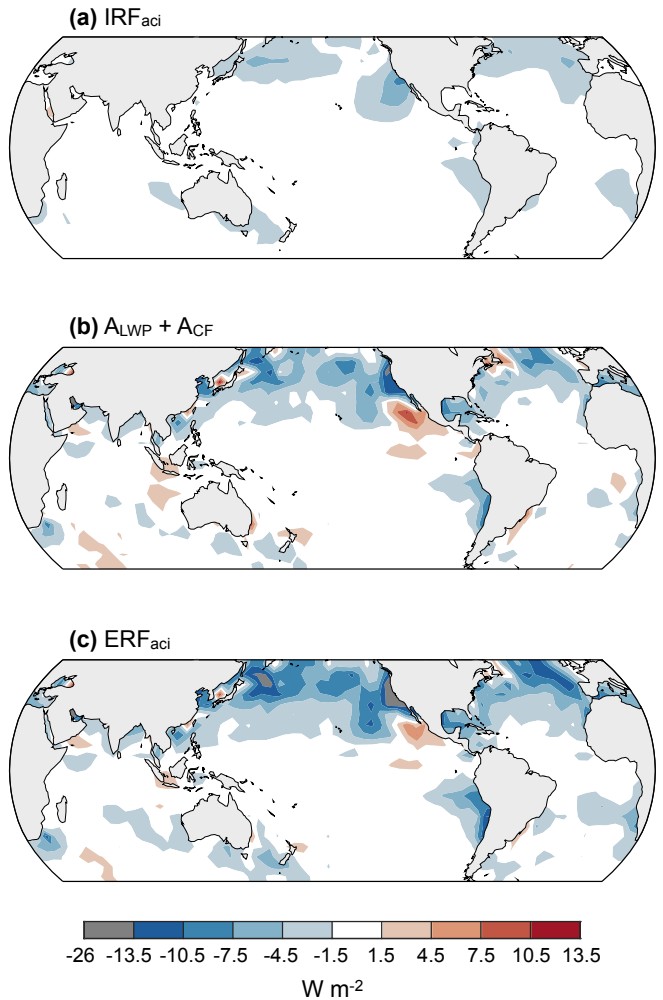

**Figure 4.** Components of historical ERF$_{aci}$ from liquid clouds, including (a) the Twomey effect (IRF$_{aci}$), (b) the total cloud adjustment (A$_{LWP}$ + A$_{CF}$), and (c) the overall ERF$_{aci}$. The estimates represent SW forcing, and they are computed with only fully cloud-covered pixels included in the cloud histograms (MODIS$_{CLD}$).

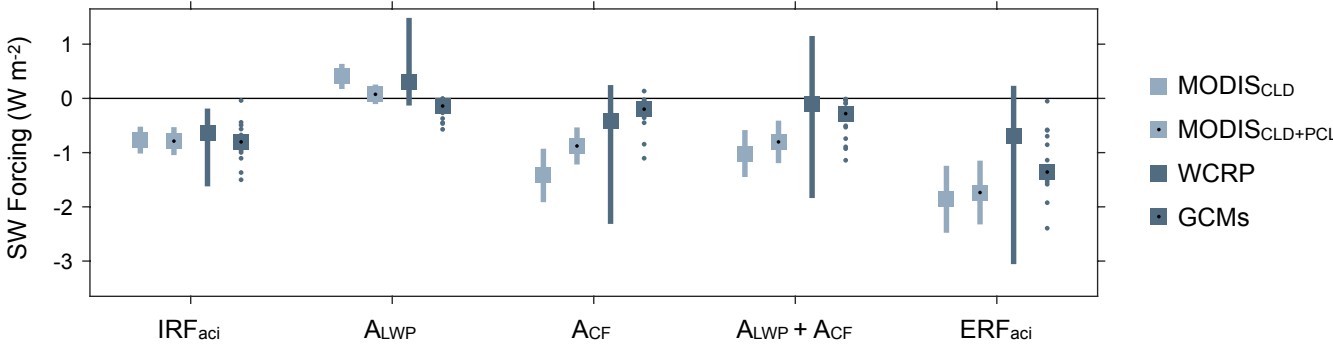

710

**Figure 5.** Components of SW ERF$_{aci}$ for liquid clouds averaged over ocean between 55°S and 55°N. The "MODIS$_{CLD}$" and "MODIS$_{CLD+PCL}$" cases show estimates from this study. The "WCRP" case is computed with the method of Bellouin et al. (2020). The "GCMs" case shows values from 14 CMIP5 and AeroCom models computed by Gryspeerdt et al. (2020). The WCRP and GCM estimates are modified from their original published values so that they represent averages over ocean between 55°S and 55°N. Squares show median values, vertical lines show 95% CIs, and dots in the "GCMs" case show individual models.

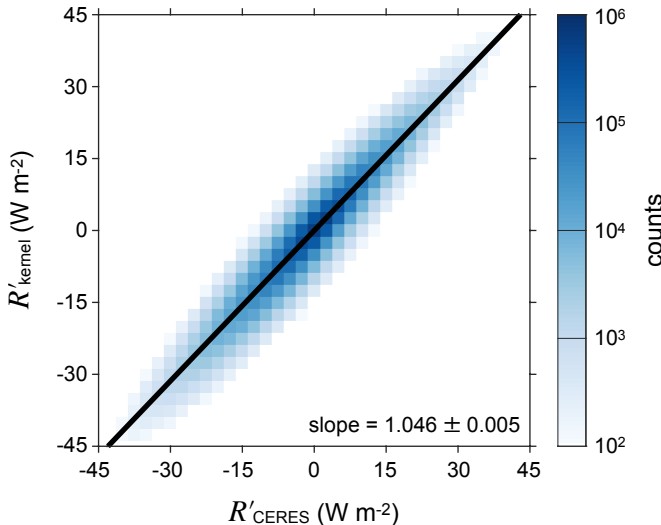

**Figure A1.** Validation of the SW cloud radiative kernel. The vertical axis shows monthly SW flux anomalies induced by liquid-topped clouds estimated with the radiative kernel and MODIS $r_e$-LWP joint histogram ($R'_{kernel}$). The horizontal axis shows monthly SW flux anomalies induced by liquid-topped clouds observed by CERES ($R'_{CERES}$). Data are plotted as a joint histogram compiled from all monthly $1° \times 1°$ grid boxes over ocean between 55°S and 55°N from 2003 through 2020. The color scale is logarithmic, and bins with fewer than 100 counts are shaded white for clarity. The black line is the ordinary least-squares regression fit. The regression slope and its 95% CI are printed in the bottom right corner.

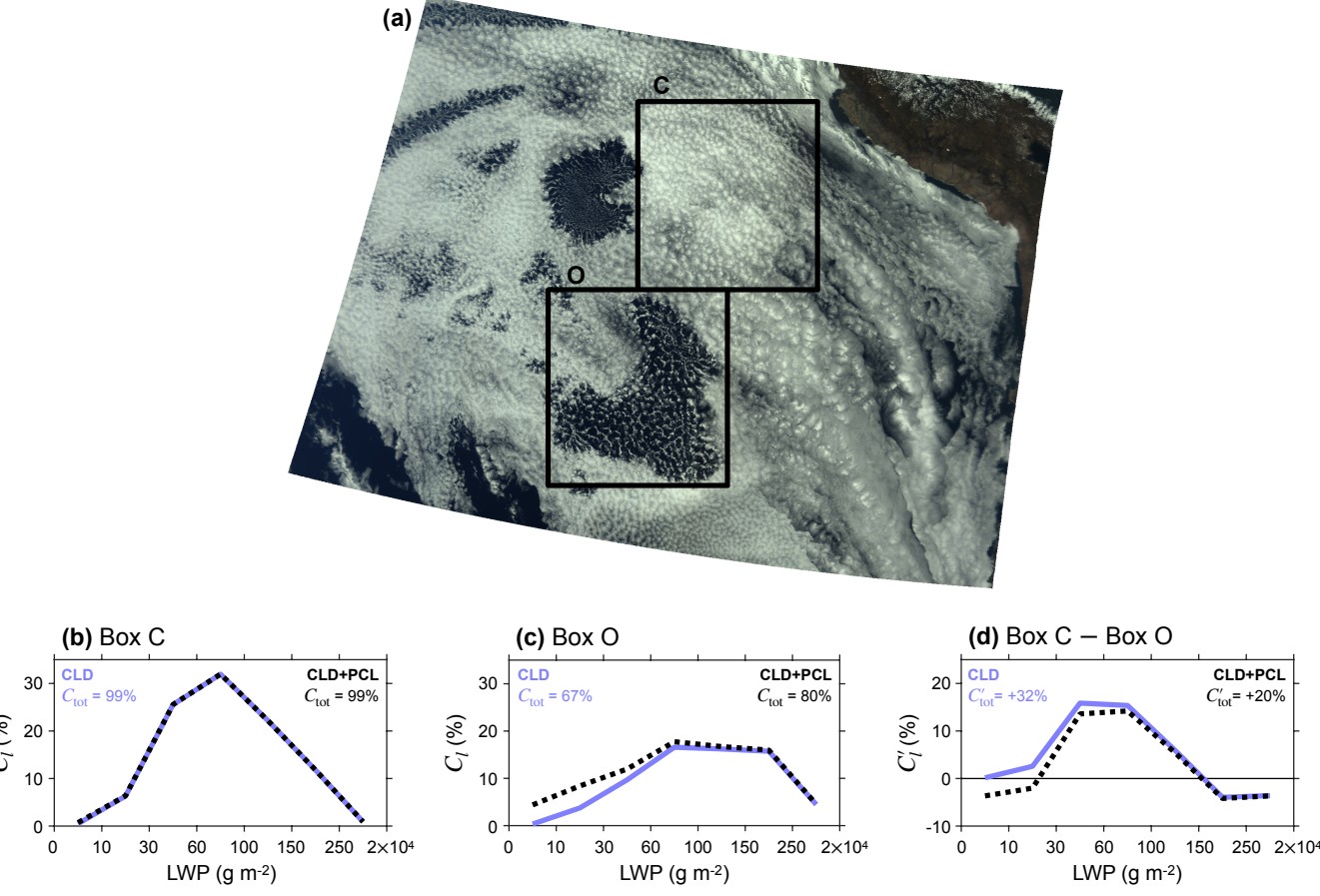

**Figure D1.** Case study demonstrating how different filtering methods for partly cloud-covered pixels lead to different estimates of LWP and cloud-fraction anomalies when the cloud morphology changes. (a) Visible image of stratocumulus clouds over the Southeast Pacific Ocean taken on September 27, 2019 from MODIS on the Terra satellite. Most clouds in box C exhibit closed mesoscale cellular convection, and most clouds in box O exhibit open mesoscale cellular convection. Both boxes span 6° latitude and 6° longitude. (b) Liquid-cloud fraction partitioned by LWP ($C_l$) in box C. In case CLD, the $C_l$ histogram includes only fully cloud-covered pixels, and in case CLD+PCL, the histogram includes both fully and partly cloud-covered pixels. The CLD and CLD+PCL cases are shown with blue and black-dashed lines, respectively. The total liquid-cloud fraction ($C_{tot}$) is printed on the figure. (c-d) Similar to (b) except that (c) shows box O and (d) shows the difference between box C and box O. Primes in (d) represent the box-C average minus the box-O average.