# Peer review of "Global Observations of Aerosol Indirect Effects from Marine Liquid Clouds"

_EGUsphere, 2023_

## Referee Comment (RC2)

**Review of "Global Observations of Aerosol Indirect Effects from Marine Liquid Clouds" by Casey J. Wall et al.**

This study quantifies SW flux perturbation arising from aerosol-cloud interactions and decomposes the flux perturbations into three components, namely the Twomey effect, LWP adjustment and CF adjustment, for global marine liquid clouds, using a combination of satellite observations, reanalysis and a radiative transfer model (RTM). These sensitivity estimates are then used to constrain ERFaci using GCMs' estimates of PI to PD aerosol changes. Their assessment framework is adopted from a previous study by the lead author (Wall et al. 2022), where TOA SW flux anomalies due to changes in marine liquid cloud properties are regressed against 7 cloud controlling factors (CCFs), including 6 key large-scale meteorological factors and one aerosol indicator, to assess the SW flux sensitivity to aerosol perturbations while controlling confounding meteorology. The innovative aspects of the current study are: i) a decomposing technique adapted from cloud-feedback literature that uses joint histograms of LWP-$r_e$, instead of COT-CTP as in Wall et al. (2022), to estimate SW flux anomalies and individual components, although an RTM is required to enable the decomposition, ii) the use of $N_d$ as an aerosol indicator, in addition to sulfate aerosol. They found that radiative forcing associated with cloud adjustments is stronger than previously believed and provided a stringent constrain on ERFaci, compared to recent assessments.

This study is no doubt publishable with profound impact and significant contributions to the ACI and ERFaci communities. The manuscript is well written and easy to follow, and the detailed documentation of methodology, framework validation, and uncertainty quantification is greatly appreciated. That said, I do have a few points that I would like the authors to consider and address first before publication.

**Key questions/concerns:**

1. Regarding the multilinear regression (MLR) method, it seems a bit concerning to learn that MLR was only able to explain less than half of the variance in R'. To me, this points to several possibilities, i) non-linear contributions from the predictors, very likely to be associated with the aerosol indicators, ii) co-linearity among CCFs, a recent work by H. Andersen et al. (2023, ACPD, the lead author of this study is a co-author) addresses this issue, iii) dependence of R' on CCFs of a larger scale, i.e. larger than 5°x5°. In other words, cloud properties may have a memory of the upstream conditions (e.g. Lewis et al. 2023 JClimate). The statistical learning framework used in Ceppi and Nowack (2021, PNAS) addressed this concern.
I wonder if the authors have taken these possibilities and methodologies into consideration, and if yes, what's the rationale to stick with the MLR framework?

2. The last condition (viii) implemented for $N_d$ selection is a bit concerning, to me at least, which essentially uses the $N_d$ from convective core regions to represent the $N_d$ of the entire cloud. This will no doubt boost the representativeness of $N_d$ on CCN, but not necessarily providing the true characterization of the $N_d$ of these clouds, as the entrainment process that also affects $N_d$ will be biasedly accounted if only convective cores are selected. I wonder how sensitive is your $\partial()/\partial(N_d)$ sensitivities to the implementation of this last condition?

If the goal is to use Nd to represent CCN as much as possible without worrying about the Nd characterization of the entire cloud, I think the reader will appreciate if this is clearly indicated in the main text.

3. I agree with the authors on the statement around lines 139-140 that, to date, no global observational studies have simultaneously estimated the three components individually. That said, there are a few that estimated intrinsic (Twomey + LWP adjustment, essentially albedo adjustment) and extrinsic sensitivities/forcings, e.g. Chen et al. (2014) and Christensen et al. (2016, 2017). More recently, Zhang & Feingold (2023, ACP) provided a bottom-up observational assessment on the temporally resolved intrinsic sensitivity (albedo susceptibility) that also controls for confounding meteorology. Since the individual components in this study is easily additive, I wonder if the authors could provide the intrinsic and extrinsic sensitivity and forcing estimates so that they can be easily compared with existing estimates?

4. Since Wall et al. (2022) is a recently published high impact study that uses a very similar assessment framework, I think it would be necessary to discuss the improvement/advantage or confidence gained by using this updated framework and reconcile the results from the previous study with that of the current study, especially the apparent stronger cloud fraction adjustment which leads to an overall more negative ERFaci (by ~0.7 W/m$^2$) in the current study.

**Other comments:**

- Lines 33-34, These cloud macrophysical adjustments have been documented in literature, please provide appropriate references.
- Line 43, it would be nice to briefly summarize what have been done along this path (i.e. existing studies/efforts on constraining ERF$_{aci}$ using satellite observations), one example is the lead author's recent study (Wall et al. 2022). What is the motivation to update the framework with this re-LWP histogram (other than enabling decomposition) and the impact of trading CERES observations with a RTM, as this is the key novelty of the current study.
- Line 154, reference?
- Lines 165-167, a bit surprising to see no indication of precip-suppression induced LWP increase, perhaps due to the cld vs cld+pcl filtering, I think the reader will appreciate some discussion along this line.
- Lines 176-177, to aggregate to global scale, don't you have to weight the regression coefficient by the frequency of occurrence of liquid cloud at each gird/location?
- Line 202, to be accurate, this assessment uses observations, reanalysis and a radiative transfer model.
- Line 267, I believe there is some sampling bias towards higher Nd, when regressing against Nd.
- Lines 327-328, Shouldn't this overestimation be taken into account when constraining ERFaci? Could you propagate this bias into your final ERFaci estimates, or, is there a reasoning for the final ERFaci estimates not being affect by this bias?
- Fig. 2-3, the unit labeling should reflect the fact that these are sensitivities values, not actual flux perturbations, i.e. unit in W m$^{-2}$ per unit increase in ln(s) or ln(Nd) (or W m$^{-2}$ ln(s/Nd)$^{-1}$).
- Please correct the year of Feingold et al. (2021, ACP) to 2022.

**Reference**

- Wall, C. J., et al.: Assessing Effective Radiative Forcing from Aerosol–Cloud Interactions over the Global Ocean. Proc. Natl. Acad. Sci. U.S.A., 119(46). https://doi.org/10.1073/pnas.2210481119, 2022.
- Andersen, H., Cermak, J., Douglas, A., Myers, T. A., Nowack, P., Stier, P., Wall, C. J., and Wilson Kemsley, S.: Sensitivities of cloud radiative effects to large-scale meteorology and aerosols from global observations, EGUsphere [preprint], https://doi.org/10.5194/egusphere-2023-1283, 2023.
- Ceppi, P. & Nowack, P. Observational evidence that cloud feedback amplifies global warming. Proc. Natl Acad. Sci. USA 118, e2026290118 (2021).
- Lewis, H., G. Bellon, and T. Dinh, 2023: Upstream Large-Scale Control of Subtropical Low-Cloud Climatology. J. Climate, 36, 3289–3303, https://doi.org/10.1175/JCLI-D-22-0676.1.
- Chen, Y.-C., Christensen, M., Stephens, G. L., and Seinfeld, J. H.: Satellite-based estimate of global aerosol–cloud radiative forcing by marine warm clouds, Nature Geosci., 7, 643–646, https://doi.org/10.1038/ngeo2214, 2014.
- Christensen, M.W., Y.-C. Chen, and G.L. Stephens, 2016a: Aerosol indirect effect dictated by liquid clouds. Journal of Geophysical Research: Atmospheres, 121(24), 14636–14650, doi:10.1002/2016jd025245.
- Christensen, M.W. et al., 2017: Unveiling aerosol–cloud interactions – Part 1: Cloud contamination in satellite products enhances the aerosol indirect forcing estimate. Atmospheric Chemistry and Physics, 17(21), 13151–13164, doi:10.5194/acp-17-13151-2017.
- Zhang, J. and Feingold, G.: Distinct regional meteorological influences on low-cloud albedo susceptibility over global marine stratocumulus regions, Atmos. Chem. Phys., 23, 1073–1090, https://doi.org/10.5194/acp-23-1073-2023, 2023.
- Feingold, G., Goren, T., and Yamaguchi, T.: Quantifying albedo susceptibility biases in shallow clouds, Atmos. Chem. Phys., 22, 3303–3319, https://doi.org/10.5194/acp-22-3303-2022, 2022.

---

## Author Response (AR1)

**Response to Reviewer Comments for "Global Observations of Aerosol Indirect Effects from Marine Liquid Clouds"**

August 3, 2023
Casey J. Wall, Trude Storelvmo, Anna Possner

We thank the reviewers for their constructive comments, which greatly improved the manuscript. Reviewer comments are written in *black italic text* below, and our responses are written in blue text.

**Comments from Ying Chen (Reviewer 1)**

*Aerosol-cloud interactions (ACI) continuously consist one of the largest uncertainties in climate radiative forcing and projections. This study combines a large ensemble of satellite observations and a statistical relation-regression method to estimate radiative forcing associated with key ACI elements, including Twomey effect, liquid water path (LWP) adjustment and cloud fraction adjustment. They found cloud fraction adjustment could be much more important than commonly believed and larger than Twomey effect in ACI cooling; while, LWP adjustment leads to slightly warming globally. The scope fit well with ACP. The manuscript is well written, the results are scientific interesting and politically meaningful supported by sound methodology. I am happy to recommend for publication after a few minor revisions.*

*Minor concerns:*
*1) Authors find the LWP adjustment leads to warming almost everywhere globally (Fig. 2); however, recent studies, which also use a large ensemble of satellite observations, reported that LWP adjustment leads to SW cooling on a large-scale (Manshausen et al., 2022;Rosenfeld et al., 2019). Could you please add some more discussion about this discrepancy?*

Thank you for suggesting this. We agree that comparing our results to other studies adds value to the paper. We added some discussion that compares our results to the findings of Diamond et al. (2020), Gryspeerdt et al. (2019), and Manshausen et al. (2022) (line 203). Regarding the Rosenfeld et al. (2019) study, there are two factors that complicate comparisons with their estimates. First, the strong SW cooling from LWP adjustments originally reported in that paper happened as a result of a coding error. When this error was corrected, they found a weak SW warming effect from the LWP adjustment, which is qualitatively consistent with our results. The coding error and updated results are described in the erratum of that paper (https://www.science.org/doi/10.1126/science.aay4194). Second, the corrected Rosenfeld et al. (2019) estimates of LWP adjustments are not reported in units of W m$^{-2}$ or fraction of the Twomey effect, so it is difficult to quantitatively compare them with our estimates. For these reasons, we chose to compare with the other three studies mentioned above.

*2) Page-3 bottom equation. Here, authors describe radiation anomaly as a function of cloud fraction (C), and the partial dependency: dR/dCrl, where C is partitioned by effective radius (r) and LWP (l). I wonder that are 'r' and 'l' the most important controlling-factors for C, or is there also other factors would largely impact 'C' and the partial dependency relationship (dR/dC)?*

The SW radiative effects of liquid clouds can be accurately predicted with Mie theory and knowledge of $r_e$, LWP, and cloud fraction, as demonstrated by the comparison between the histogram/kernel predictions and the CERES observations in Fig. A1. Variations in cloud-top pressure can affect liquid-cloud radiative effects by changing the amount of SW radiation that is absorbed by atmospheric gases and aerosols above the cloud top, but this effect is small relative to the radiative anomalies associated with typical variations in $r_e$, LWP, and cloud fraction. For ice clouds, variations in particle shape and surface roughness can change SW cloud radiative effects by modifying the scattering phase function and single-scattering albedo, but these factors are not relevant to liquid clouds.

*3) Some more clarification about the method would help audience better understand it. A) line-100 (and after), what does 'anomaly' here refer to, do you mean anomaly to the climatological value (temporal averaged, or also spatial averaged)? B) line-120: some description about how do you remove the climatological seasonal cycle and linear trend. C) line-125: explain 46-49% variance – how do you measure variance and lead to this conclusion?*

(A) We compute the climatological seasonal cycle separately at each latitude-longitude grid-box by averaging the data over all of the January data points, averaging the data over all of the February data points, and so on. We changed the text to emphasize that we compute the climatological seasonal cycle at each grid box rather than computing the spatially averaged climatological seasonal cycle (line 100, 133, and 371).

(B) We changed "linear trend" to "least-squares linear trend" to clarify how it is calculated (line 133).

(C) To determine the goodness of fit of the regression model, we compute the fraction of $R'$ variance that it explains in each latitude-longitude grid-box from the standard coefficient of determination, then spatially average the results over the domain. We added a sentence that states this (line 137).

*4) Fig. 1. Joint histogram. I do not quite understand this figure. Does the color indicate the values of cloud fraction (a) and SW kernel (b)? If yes, then this is not a joint histogram, it is a heatmap plot. A histogram should show the probability density function (or counts) of data.*
*Moreover, Fig. 1b. the kernel dR/dC should be depended on latitude/longitude/day-of-the-year/surface-albedo. Does all of these factors are controlled, e.g., fixed to an average value, and only allow re and LWP to vary?*

The standard MODIS joint histograms are formatted in units of pixel counts. We normalize the pixel counts by the number of all valid pixels in the grid box and then multiply by 100 to convert the units to cloud fraction. We modified the text to clarify this (line 57). We prefer to use the terminology "joint histogram" because this is standard

practice in the cloud-feedback literature from which our method was adapted (Zelinka et al., 2012), and we want to streamline comparisons between our work and that body of literature.

Regarding the kernel, it is a function of latitude, longitude, calendar month, LWP, and $r_e$ (line 356). Fig. 1b shows the kernel averaged over the latitude, longitude, and time dimensions. We changed the caption of Fig. 1 to clarify this.

*5) Data open-access. SW kernel data is a key factor use in this study and generated in this study. I feel that making the global distribution of this dataset open-access would largely improve the reproducibility of this study, and also enhance its contribution to the community.*

We posted the cloud radiative kernel for public access on GitHub and added the website link to the Data Availability section.

**Comments from Jianhao Zhang (Reviewer 2)**
*This study quantifies SW flux perturbation arising from aerosol-cloud interactions and decomposes the flux perturbations into three components, namely the Twomey effect, LWP adjustment and CF adjustment, for global marine liquid clouds, using a combination of satellite observations, reanalysis and a radiative transfer model (RTM). These sensitivity estimates are then used to constrain ERFaci using GCMs' estimates of PI to PD aerosol changes. Their assessment framework is adopted from a previous study by the lead author (Wall et al. 2022), where TOA SW flux anomalies due to changes in marine liquid cloud properties are regressed against 7 cloud controlling factors (CCFs), including 6 key large-scale meteorological factors and one aerosol indicator, to assess the SW flux sensitivity to aerosol perturbations while controlling confounding meteorology. The innovative aspects of the current study are: i) a decomposing technique adapted from cloud-feedback literature that uses joint histograms of LWP-re, instead of COT-CTP as in Wall et al. (2022), to estimate SW flux anomalies and individual components, although an RTM is required to enable the decomposition, ii) the use of Nd as an aerosol indicator, in addition to sulfate aerosol. They found that radiative forcing associated with cloud adjustments is stronger than previously believed and provided a stringent constrain on ERFaci, compared to recent assessments.*

*This study is no doubt publishable with profound impact and significant contributions to the ACI and ERFaci communities. The manuscript is well written and easy to follow, and the detailed documentation of methodology, framework validation, and uncertainty quantification is greatly appreciated. That said, I do have a few points that I would like the authors to consider and address first before publication.*

*Key questions/concerns:*
*1. Regarding the multilinear regression (MLR) method, it seems a bit concerning to learn that MLR was only able to explain less than half of the variance in R'. To me, this points to several possibilities, i) non-linear contributions from the predictors, very likely to be associated with the aerosol indicators, ii) co-linearity among CCFs, a recent work*

*by H. Andersen et al. (2023, ACPD, the lead author of this study is a co-author) addresses this issue, iii) dependence of R' on CCFs of a larger scale, i.e. larger than 5ox5o . In other words, cloud properties may have a memory of the upstream conditions (e.g. Lewis et al. 2023 JClimate). The statistical learning framework used in Ceppi and Nowack (2021, PNAS) addressed this concern. I wonder if the authors have taken these possibilities and methodologies into consideration, and if yes, what's the rationale to stick with the MLR framework?*

Wall et al. (2022; hereafter W22) showed that the MLR model used in this study can accurately predict the regional cloud response in several cases with known variations in aerosol sources, including decadal cloud trends downwind of major emission sources in North America, decadal cloud trends downwind of major emission sources in Asia, and the cloud response to effusive eruptions of Kilauea Volcano in Hawaii. We believe that these findings justify the use of the MLR model for estimating $ERF_{aci}$. Of course, it is possible that a more complex statistical model that includes non-local or non-linear relationships between clouds and their controlling factors could explain more of the $R'$ variance and thus provide more precise predictions of aerosol indirect effects. We modified the conclusion to state that investigating non-linear and non-local relationships between clouds and their controlling factors could be a way to improve upon our results (line 306).

Regarding collinearity between predictor variables, the standard MLR model used in this study accounts for collinearity when quantifying regression-coefficient uncertainty (O'Brien, 2007). In general, it is possible for collinearity to cause the regression coefficients to be too uncertain to obtain useful statistical results, but the existence of collinearity does not invalidate the MLR model because collinearity is accounted for when quantifying regression-coefficient uncertainty.

*2. The last condition (viii) implemented for Nd selection is a bit concerning, to me at least, which essentially uses the Nd from convective core regions to represent the Nd of the entire cloud. This will no doubt boost the representativeness of Nd on CCN, but not necessarily providing the true characterization of the Nd of these clouds, as the entrainment process that also affects Nd will be biasedly accounted if only convective cores are selected. I wonder how sensitive is your partial()/partial(Nd) sensitivities to the implementation of this last condition? If the goal is to use Nd to represent CCN as much as possible without worrying about the Nd characterization of the entire cloud, I think the reader will appreciate if this is clearly indicated in the main text.*

Indeed, the goal is to use the $N_d$ in cloud elements with the largest 10% optical thickness (convective-core $N_d$) as an indicator of cloud-base CCN concentration rather than to characterize $N_d$ in the entire cloud. We added a sentence to emphasize this (line 76), and we now refer to the filtered cloud-droplet number concentration as $\widetilde{N}_d$ to distinguish it from the cloud-droplet number concentration in the entire cloud (line 78). The caption of Fig. 2 also states the filtering condition to remind the reader. Supplementary Fig. 2 shows that the results are qualitatively consistent when filtering methods for $N_d$ recommended by Grosvenor et al. (2018) and Bennartz and Rausch (2017) are used instead of the largest 10% cloud optical thickness values (line 211).

*3. I agree with the authors on the statement around lines 139-140 that, to date, no global observational studies have simultaneously estimated the three components individually. That said, there are a few that estimated intrinsic (Twomey + LWP adjustment, essentially albedo adjustment) and extrinsic sensitivities/forcings, e.g. Chen et al. (2014) and Christensen et al. (2016, 2017). More recently, Zhang & Feingold (2023, ACP) provided a bottom-up observational assessment on the temporally resolved intrinsic sensitivity (albedo susceptibility) that also controls for confounding meteorology. Since the individual components in this study is easily additive, I wonder if the authors could provide the intrinsic and extrinsic sensitivity and forcing estimates so that they can be easily compared with existing estimates?*

W22 computed terms that closely approximate the "intrinsic" and "extrinsic" components of aerosol indirect effects for low-level clouds, although they did not use that terminology. Unfortunately, they could not reach strong conclusions from these terms because they depend substantially on whether partly cloudy pixels are filtered out or included in the cloud histograms – as documented for some other terms in the current study as well. Discussing the "intrinsic" component would be very similar to what W22 did, so we prefer to focus the discussion on what is new in the current paper instead. The "extrinsic" component is equivalent to the estimated cloud-fraction adjustment in the present study, which is discussed in the current paper. For these reasons, we prefer not to discuss "intrinsic" and "extrinsic" components of aerosol indirect effects in the main text.

That said, the reviewer makes an excellent point that stating the values for the components of aerosol indirect effects, and different combinations of the components, would help to facilitate comparisons with other studies and support efforts to reproduce our results. We therefore added a table to the Supplementary Information that lists estimates of all of the components of aerosol indirect effects and $ERF_{aci}$, including the "intrinsic" and "extrinsic" components (Supplementary Table 3). This table is referenced in the Data Availability section.

*4. Since Wall et al. (2022) is a recently published high impact study that uses a very similar assessment framework, I think it would be necessary to discuss the improvement/advantage or confidence gained by using this updated framework and reconcile the results from the previous study with that of the current study, especially the apparent stronger cloud fraction adjustment which leads to an overall more negative ERFaci (by ~0.7 W/m2 ) in the current study.*

There are three main factors that cause the difference in the estimated $ERF_{aci}$ between the current study and W22. First, the current study estimates $ERF_{aci}$ for all liquid-topped clouds, while W22 estimate $ERF_{aci}$ for low-level clouds, defined as clouds with tops between the surface and 680 hPa. The current study thus includes a larger subset of the overall cloud population. We applied the method of W22 to estimate the SW $ERF_{aci}$ for all liquid-topped clouds, and we found that the result is about 26% larger in magnitude than the estimate of SW $ERF_{aci}$ from low-level clouds. Second, the current study estimates SW $ERF_{aci}$, while W22 estimate net $ERF_{aci}$. Thus, their estimate includes an additional $ERF_{aci}$ component from changes in longwave radiation, which offsets about 14% of the SW component (Appendix B). Third, the current study estimates $ERF_{aci}$ with MODIS data and radiative kernels, while W22 estimate $ERF_{aci}$

with CERES data for their main result. The MODIS data and kernels in the current study overestimate the magnitude of SW cloud radiative effects relative to CERES observations by about 4.6% (Fig. A1). These three factors cause the estimates in the current study to have a larger magnitude than those reported by W22. Considering these factors and the uncertainties in the ERF$_{aci}$ estimates of both studies, our estimates are consistent with those of W22.

We added a paragraph that explains all of these methodological differences, and we moved the information to the section on historical aerosol forcing so that it is easier for the reader to remember when the ERF$_{aci}$ values are presented (line 236).

*Other comments:*
▪ Lines 33-34, These cloud macrophysical adjustments have been documented in literature, please provide appropriate references.
We added four key citations: Albrecht (1989), Pincus and Baker (1994), Rosenfeld et al. (2006), and Bretherton et al. (2007).

▪ Line 43, it would be nice to briefly summarize what have been done along this path (i.e. existing studies/efforts on constraining ERFaci using satellite observations), one example is the lead author's recent study (Wall et al. 2022). What is the motivation to update the framework with this re-LWP histogram (other than enabling decomposition) and the impact of trading CERES observations with a RTM, as this is the key novelty of the current study.
The main novelty of the current study is that it develops a single, self-consistent framework to simultaneously estimate the Twomey effect, LWP adjustment, and cloud-fraction adjustment at a near-global scale. We changed the text to emphasize this (line 46).

▪ Line 154, reference?
We originally wrote this introductory sentence without a reference because we wanted to provide more specific details and relevant references in the following sentences of the paragraph. We added a reference to Zhang et al. (2022) since it concisely shows observational support for regime-dependent cloud adjustments.

▪ Lines 165-167, a bit surprising to see no indication of precip-suppression induced LWP increase, perhaps due to the cld vs cld+pcl filtering, I think the reader will appreciate some discussion along this line.
We added some discussion about relationships between aerosols, precipitation occurrence, and LWP adjustments (line 181).

▪ Lines 176-177, to aggregate to global scale, don't you have to weight the regression coefficient by the frequency of occurrence of liquid cloud at each gird/location?
The regression coefficients for $\partial R/\partial \ln s$ and $\partial R/\partial \ln \widetilde{N}_d$ (and their components) have units of W m$^{-2}$ averaged over the entire grid box, so it is appropriate to weight them by grid-box area when aggregating to the global scale. If one were to examine variables that are averaged over the liquid-cloud-covered area inside grid boxes instead, then it would be appropriate to weight them by liquid-cloud area when aggregating to the

global scale. The current study does not examine any variables that are averaged over the liquid-cloud-covered area inside grid boxes, so the second case does not apply to our analysis.

▪ Line 202, to be accurate, this assessment uses observations, reanalysis and a radiative transfer model.
We changed this sentence to state that our method uses "observations, reanalysis, and radiative transfer modeling" (line 226).

▪ Line 267, I believe there is some sampling bias towards higher Nd, when regressing against Nd.
We found qualitatively consistent results for the relative importance of the Twomey effect and cloud adjustments when using $\widetilde{N}_d$ and sulfate concentration as the indicator for cloud-base CCN concentration (Fig. 2, 3). This shows that the main results are robust between these two independent CCN indicators. Thus, we agree that sampling biases are possible for $N_d$, but this potential limitation does not affect the main interpretation.

▪ Lines 327-328, Shouldn't this overestimation be taken into account when constraining ERFaci? Could you propagate this bias into your final ERFaci estimates, or, is there a reasoning for the final ERFaci estimates not being affect by this bias?
We use CERES observations as "ground truth" for quantifying bias of the MODIS/kernel estimates of cloud radiative effects. Because of the formatting of the CERES data, we can compute the bias of the MODIS/kernel estimates of $R'$, but we cannot compute the bias of the estimates of $R'_{r_e}$, $R'_{\mathrm{LWP}}$, or $R'_{\mathrm{CF}}$. Thus, it is possible to bias-correct the overall ERF$_{\mathrm{aci}}$, but not the Twomey effect or adjustment components. We think it would be too confusing to bias-correct the overall ERF$_{\mathrm{aci}}$ but not the other components since the components would no longer add up to the overall ERF$_{\mathrm{aci}}$. We believe that this is justified because the bias of $R'$ (+4.6%) is much smaller than the uncertainty range for the overall ERF$_{\mathrm{aci}}$ ($\pm 33\%$). However, we want to make sure that this limitation is clear to the reader, so we now state the bias of MODIS/kernel estimates of $R'$ in the main text to make this information more visible (line 114 and line 243).

▪ Fig. 2-3, the unit labeling should reflect the fact that these are sensitivities values, not actual flux perturbations, i.e. unit in W m-2 per unit increase in ln(s) or ln(Nd) (or W m-2 ln(s/Nd)-1 ).
Anomalies of $R$ have units of W m$^{-2}$, and anomalies of $\ln s$ and $\ln N_d$ are unitless because they represent fractional changes in $s$ and $N_d$:
$$\ln(s + \Delta s) - \ln(s) = \ln\left(\frac{s + \Delta s}{s}\right) = \ln\left(1 + \frac{\Delta s}{s}\right) \approx \frac{\Delta s}{s}$$
and similarly for $N_d$. Thus, the units of the regression coefficients representing $\partial R / \partial \ln s$ and $\partial R / \ln N_d$ are W m$^{-2}$.

▪ Please correct the year of Feingold et al. (2021, ACP) to 2022.
Thank you for catching this! We corrected it.

**References**

Albrecht, B. A.: Aerosols, cloud microphysics, and fractional cloudiness. *Science*, *245*(4923), 1227–1230. https://doi.org/10.1126/science.245.4923.1227, 1989.

Bennartz, R., and Rausch, J.: Global and Regional Estimates of Warm Cloud Droplet Number Concentration Based on 13 Years of AQUA-MODIS Observations, *Atmos. Chem. Phys.*, 17, 9815– 9836, https://doi.org/10.5194/acp-17-9815-2017, 2017.

Bretherton, C. S., Blossey, P. N., and Uchida, J.: Cloud Droplet Sedimentation, Entrainment Efficiency, and Subtropical Stratocumulus Albedo. *Geophys. Res. Lett.*, *34*(3), 1–5. https://doi.org/10.1029/2006GL027648, 2007.

Diamond, M. S., et al.: Substantial Cloud Brightening From Shipping in Subtropical Low Clouds. *AGU Adv.*, *1*(1), 1–28. https://doi.org/10.1029/2019av000111, 2020.

Grosvenor, D. P., et al.: Remote Sensing of Droplet Number Concentration in Warm Clouds: A Review of the Current State of Knowledge and Perspectives. *Rev. Geophy.*, *56*(2), 409–453. https://doi.org/10.1029/2017RG000593, 2018.

Gryspeerdt, E., et al.: Constraining the Aerosol Influence on Cloud Liquid Water Path. *Atmos. Chem. Phys.*, *19*(8), 5331–5347. https://doi.org/10.5194/acp-19-5331-2019, 2019.

Manshausen, P., et al.: Invisible ship tracks show large cloud sensitivity to aerosol. *Nature*, *610*(7930), 101–106. https://doi.org/10.1038/s41586-022-05122-0, 2022.

O'Brien, R. M. A caution regarding rules of thumb for variance inflation factors. *Quality and Quantity*, *41*(5), 673–690. https://doi.org/10.1007/s11135-006-9018-6, 2007.

Pincus, R., and Baker, M. B.: Effect of Precipitation on the Albedo Susceptibility of Clouds in the Marine Boundary Layer. *Nature*, *372*(6503), 250–252. https://doi.org/10.1038/372250a0, 1994.

Rosenfeld, D., Kaufman, Y. J., and Koren, I.: Switching Cloud Cover and Dynamical Regimes from Open to Closed Benard Cells in Response to the Suppression of Precipitation by Aerosols. *Atmos. Chem. Phys.*, *6*(9), 2503–2511. https://doi.org/10.5194/acp-6-2503-2006, 2006.

Rosenfeld, D., et al.: Aerosol-Driven Droplet Concentrations Dominate Coverage and Water of Oceanic Low-Level Clouds. *Science*, *363*(6427). https://doi.org/10.1126/science. aav0566, 2019.

Wall, C. J., et al.: Assessing Effective Radiative Forcing from Aerosol–Cloud Interactions over the Global Ocean. *Proc. Natl. Acad. Sci. U.S.A.*, *119*(46). https://doi.org/10.1073/pnas. 2210481119, 2022.

Zelinka, M. D., Klein, S. A., and Hartmann, D. L.: Computing and Partitioning Cloud Feedbacks Using Cloud Property Histograms. Part I: Cloud Radiative Kernels. *J. Clim.*, *25*(11), 3715–3735. https://doi.org/10.1175/JCLI-D-11-00248.1, 2012.

Zhang, J., Zhou, X., Goren, T., and Feingold, G.: Albedo susceptibility of northeastern Pacific stratocumulus: the role of covarying meteorological conditions, Atmos. Chem. Phys., 22, 861–880, https://doi.org/10.5194/acp-22-861-2022, 2022.